# Multimodal analysis of cell-free DNA whole-methylome sequencing for cancer detection and localization

Fenglong Bie [1,2,6], Zhijie Wang[3,6], Yulong Li [4,6], Wei Guo [1,6], Yuanyuan Hong[4], Tiancheng Han[4], Fang Lv[4], Shunli Yang[4], Suxing Li[4], Xi Li[4], Peiyao Nie[4], Shun Xu[5], Ruochuan Zang[1], Moyan Zhang[1], Peng Song[1], Feiyue Feng[1], Jianchun Duan[3], Guangyu Bai[1], Yuan Li[1], Qilin Huai[1], Bolun Zhou[1], Yu S. Huang [4], Weizhi Chen[4], Fengwei Tan [1] ✉ & Shugeng Gao [1] ✉

Multimodal epigenetic characterization of cell-free DNA (cfDNA) could improve the performance of blood-based early cancer detection. However, integrative profiling of cfDNA methylome and fragmentome has been technologically challenging. Here, we adapt an enzyme-mediated methylation sequencing method for comprehensive analysis of genome-wide cfDNA methylation, fragmentation, and copy number alteration (CNA) characteristics for enhanced cancer detection. We apply this method to plasma samples of 497 healthy controls and 780 patients of seven cancer types and develop an ensemble classifier by incorporating methylation, fragmentation, and CNA features. In the test cohort, our approach achieves an area under the curve value of 0.966 for overall cancer detection. Detection sensitivity for early-stage patients achieves 73% at 99% specificity. Finally, we demonstrate the feasibility to accurately localize the origin of cancer signals with combined methylation and fragmentation profiling of tissue-specific accessible chromatin regions. Overall, this proof-of-concept study provides a technical platform to utilize multimodal cfDNA features for improved cancer detection.

Cancer is becoming the most deadly disease, accounting for almost 10 million deaths around the globe in 2020[1]. Detecting cancer early provides an opportunity for more effective therapeutic intervention, which may reduce treatment morbidity and mortality. In recent years, cfDNA-based liquid biopsy has gained prominent interest in early cancer detection and diagnosis with its minimal invasiveness and the potential to reveal tiny tumors. However, tumor-originated cfDNA, or circulating tumor DNA (ctDNA), constitutes only a small fraction of cfDNA[2,3]. The advancement of experimental and computational methodologies to improve the signal-to-noise ratio for more sensitive cancer detection remains a constant clinical need.

While genetic mutation is a hallmark of cancer[4], plasma mutations are challenging to detect given the low fraction of mutation-bearing ctDNA fragments in early-stage disease or certain tumor types[5,6]. There is also increasing evidence for the presence of somatic mutations in non-malignant tissues[7,8], which may hamper the specificity of mutation search for cancer detection. In contrast, epigenetic dysregulation is an early-occurring event in tumorigenesis and involves widespread

[1]Department of Thoracic Surgery, National Cancer Center/National Clinical Research Center for Cancer/Cancer Hospital, Chinese Academy of Medical Sciences and Peking Union Medical College, Beijing 100021, China. [2]Department of Thoracic Surgery, Shandong Provincial Hospital Affiliated to Shandong First Medical University, Jinan 250021 Shandong, China. [3]Department of Medical Oncology, National Cancer Center/National Clinical Research Center for Cancer/Cancer Hospital, Chinese Academy of Medical Sciences and Peking Union Medical College, Beijing 100021, China. [4]Genecast Biotechnology Co., Ltd., Wuxi 214105 Jiangsu, China. [5]Department of Thoracic Surgery, The First Hospital of China Medical University, Shenyang, Liaoning Province 110001, China. [6]These authors contributed equally: Fenglong Bie, Zhijie Wang, Yulong Li, Wei Guo. ✉e-mail: tanfengwei@cicams.ac.cn; gaoshugeng@cicams.ac.cn

alterations of DNA methylation and chromatin organization in both cancer cells and tumor microenvironment[9,10]. Compared with searching point mutations, characterizing the vast number of plasma epigenetic changes is expected to improve detection sensitivity, as shown by recent studies about the promise of cfDNA methylation profiling for cancer detection and localization[11,12]. However, conventional bisulfite-based methylation sequencing severely damages DNA[13], which is both material consuming and inapplicable for profiling other informative cfDNA features such as fragmentation.

Recently, a novel bisulfite-free method utilizing mild TET2 and APOBEC3A enzymes was introduced for methylation detection of genomic DNA with reduced DNA damage[14]. This nondestructive nature opens up a potential avenue for simultaneous methylation and fragmentation analysis for enhanced cancer detection, although its utility in cfDNA sequencing remains to be explored. Here, we adapted this method for whole-methylome sequencing (WMS) of cfDNA extracted from only 4 ml of plasma and demonstrated proof that cfDNA WMS is highly concordant with whole-genome sequencing (WGS) in fragmentation and coverage profiling. Moreover, all these genomic features were readily detectable at low sequencing depth and thus reducing sequencing cost.

To develop and validate a multicancer detection test, we applied shallow WMS to plasma samples from a multicenter, case-control, observational MONITOR (Multi-Omics Noninvasive Inspection of TumOr Risk) study comprising seven common cancer types and healthy controls (Supplementary Data 1). We developed computational methods to extract four types of cancer-associated features across the plasma genome, including methylation[15], fragment size[16], copy number alteration[17], and fragment end motif[18], and constructed an ensemble machine learning classifier integrating all modalities for distinguishing cancer patients from healthy controls with high sensitivity across stages. By combinatorial analysis of methylation and fragmentation signatures at cancer tissue-specific accessible chromatin regions, we were able to accurately locate the tissue origin of cancer signals. Overall, this proof-of-concept study provides a blood-based approach, termed THEMIS (THorough Epigenetic Marker Integration Solution), for sensitive and accurate multicancer early detection and localization.

## Results

### Overview of THEMIS approach for cancer detection

Figure 1 illustrates the experimental and computational workflow of cancer detection by THEMIS approach. Unlike bisulfite sequencing, we adapted an enzyme-based method[14] to characterize the whole-genome methylome of cfDNA extracted from 4 ml of plasma. This method utilizes TET2 to protect methylcytosines from subsequent deamination by APOBEC3A, which converts unmodified cytosines to uracils. We spiked-in unmethylated lambda DNA to estimate the conversion rate of unmodified cytosine, and the 1,277 cfDNA samples included in the MONITOR cohort had a median conversion rate of 99.4% (Fig. S1). Although this method detects methylation at single-base resolution, we subjected WMS libraries to low-pass paired-end sequencing to limit sequencing cost after determining the minimum required sequencing depth in our pilot analysis. All uniquely aligned sequencing data were randomly downsampled to 60 million properly paired reads (~2X haploid genome coverage) for downstream analysis and model development.

Given that the mild enzymatic reactions minimize DNA damage, we sought to incorporate fragmentation and CNA features in addition to methylation for enhanced cancer detection. We designed algorithms to extract four epigenetic and genetic cancer features from plasma cfDNA WMS data. To profile the genome-wide methylation patterns, we divided the genome into 1846 nonoverlapping 1-Mb segments and calculated the ratio of fully methylated fragments within each window (Methylated Fragment Ratio, MFR). Similarly, position-specific fragmentation characteristics were profiled as the ratios of short (100–166 bp) to long (169–240 bp) fragments for 502 non-overlapping 5-Mb genomic windows (Fragment Size Index, FSI). To enhance the signal of copy number alteration, we size-selected short (<151 bp) and long (>220 bp) fragments which are more likely to originate from cancer cells[19,20] to quantify the copy number changes of chromosome arms (Chromosomal Aneuploidy of Featured Fragments, CAFF). In addition, the frequencies of 256 4-mer motifs at the 5' end of fragments were quantified (Fragment End Motif, FEM). To improve the performance of cancer detection, machine learning methods were implemented for MFR, FSI, and FEM modalities, including support vector machine (SVM) models for MFR and FSI, and a logistic regression (LR) model for FEM, after reduction of data dimensionality by principal component analysis (PCA). Finally, an ensemble classifier (THEMIS) was constructed using a regularized LR model to integrate prediction results from the four individual modalities. We used the output of the THEMIS model, which was termed the THEMIS score, to predict a sample's probability of having cancer,

### Copy number and fragmentation profiling from cfDNA WMS

A major advantage of WMS over whole-genome bisulfite sequencing (WGBS) is the ability to keep most DNA intact, which enables the extraction of additional genetic and fragmentation information to improve the sensitivity of cancer detection. We included a cohort of 220 healthy controls and 270 cancer patients of multiple types (Supplementary Data 2) to evaluate copy number and fragmentation profiling from cfDNA WMS data and assessed their concordance with whole-genome sequencing (WGS) data of similar sequencing depth (60M reads) generated with the same cfDNA samples. We first investigated the similarity of cfDNA copy number landscape between WMS and WGS data. As observed for the plasma genome of an advanced colorectal cancer (COREAD) patient GCP0088 (Fig. 2a) as well as examples of other cancer types (Fig. S2a), WMS displayed highly consistent copy number patterns with WGS in genomic bins of 100 kb. We noted the same chromosome alterations frequently observed in colorectal cancer, including gains of chromosomes 13 and 20 and losses of chromosomes 4 and 18[21], with both sequencing platforms (Fig. 2a). We used a metric called plasma aneuploidy score (PA score) which summarized copy number changes of the top five chromosome arms[17] to evaluate the aneuploidy level of a sample. Within the entire cohort samples, PA scores were closely matched between WMS and WGS data with a Pearson correlation coefficient (PCC) of 0.988 (95% confidence interval (CI): 0.986–0.990) (Fig. 2b). These data confirm the concordance between WMS and WGS in detecting cfDNA CNA.

Next, we investigated whether WMS could retain cfDNA fragmentation patterns, which displayed position-specific changes in the distribution of fragment size in cancer patient plasma[16,22]. We defined the coverage ratio of short to long fragments in 5-Mb windows along the genome as fragmentation size index (FSI). Across the plasma genome of Patient GCP0088 as well as patients of other cancer types, similar fragmentation profiles were observed between WMS and WGS data (Figs. 2c and S2b). Because cfDNA fragmentation profiles among healthy individuals were highly consistent[16], for each platform we generated a reference FSI profile from healthy controls by taking the median FSI value of each window. We then measured the similarity of each sample's FSI profile to the reference profile with PCC as a proxy for fragmentation pattern. Among all cohort samples, WMS PCCs and WGS PCCs showed high concordance with a correlation coefficient of 0.961 (95% CI: 0.954–0.968) (Fig. 2d). These data demonstrate that WMS can retain faithful cfDNA fragmentation signature.

### Plasma methylation, fragmentation, and CNA profiles provide complementary cancer-associated signals

The multimodal genomic features we profiled may represent different molecular alterations underlying tumorigenesis[23]. To investigate

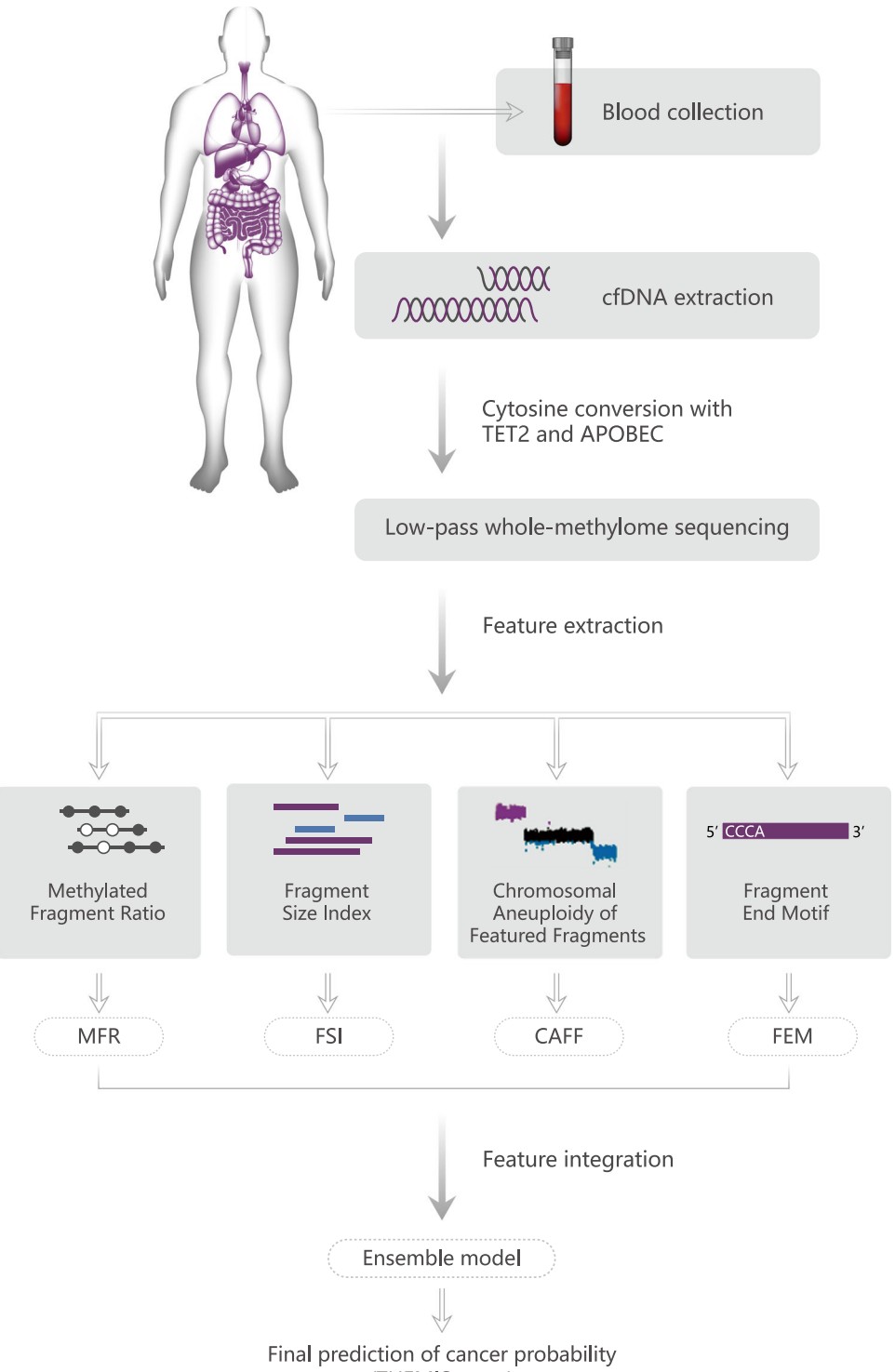

**Fig. 1 | Overview of THEMIS approach for cancer detection based on plasma whole-methylome sequencing.** Schematic illustration of the experimental and bioinformatics procedures. Blood samples were collected from cancer patients or noncancer control donors. Plasma cfDNA was extracted from the participant's blood sample and subject to low-pass WMS using TET2 and APOBEC enzymes for cytosine conversion. Four modalities were extracted from uniquely mapped WMS sequencing reads, including Methylated Fragment Ratio (MFR), Fragment Size Index (FSI), Chromosomal Aneuploidy of Featured Fragments (CAFF), and Fragment End Motif (FEM). An ensemble model integrating prediction scores from all four modalities was constructed to yield the final probability of having cancer, termed THEMIS score, for a sample.

whether these features could provide complementary signals in cancer detection, we analyzed the associations among fragmentation (FSI), methylation (MFR), and CNA profiles of 1,795 1-Mb nonoverlapping windows along the plasma genome of 20 healthy controls and 20 colorectal cancer samples (Supplementary Data 3). For each feature, z-score over healthy controls was shown for each window after quantile-normalization with healthy controls (Fig. 3a–c). Compared with healthy controls, cancer patient plasma genome exhibited both increases and decreases in the signals of all three features spanning broad regions. We then analyzed the number of commonly altered

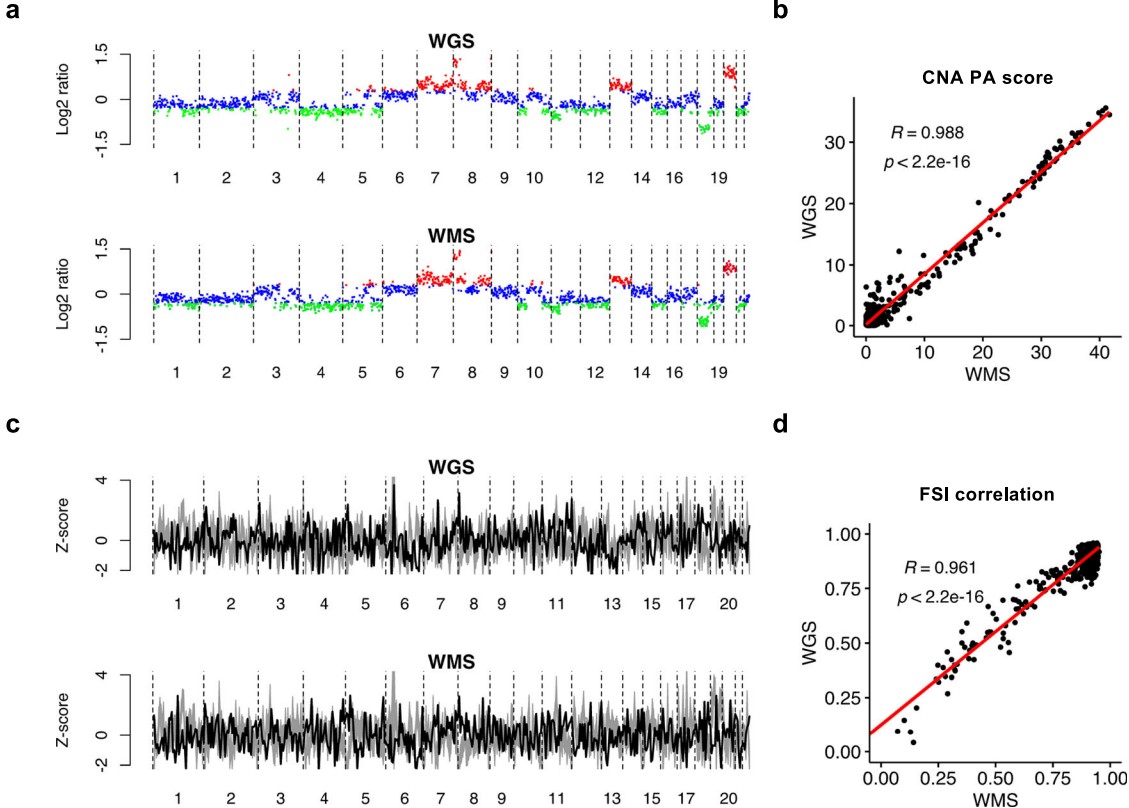

**Fig. 2 | Concordance between WMS and WGS in cfDNA copy number and fragmentation profiling. a**, **c** depict the genome-wide CNA and FSI profiles of a colorectal cancer patient GCP0088 as an example. **b**, **d** analyze 225 healthy controls and 287 cancer patients for cohort-level comparison. **a** Log2 ratio over the mean baseline coverage in 100-kb bins across the genome of Patient GCP0088 profiled by WGS and WMS respectively. Bins with log2 ratio above 0.3 (copy number gains) are colored in red, and bins with log2 ratio below −0.3 (copy number loss) are colored in green. **b** Scatter plot of PA scores derived from matched WGS and WMS data of all samples. The regression line is colored in red. Pearson correlation coefficient ($R$) of 0.988 (95% confidence interval: 0.986–0.990) and the $p$ value (<2.2 × 10⁻⁶,

two-sided) are indicated on the plot. **c** FSI of 5-Mb windows across the genome of Patient GCP0088 profiled by WGS and WMS respectively. The coverage ratio of short to long fragments for each bin is normalized by z-score across the genome. Patient FSI profile is colored in black against 225 gray healthy reference FSI profiles. **d** Scatter plot of Pearson correlations between FSI profiles of individual samples with the reference FSI profile, assessed with WGS and WMS data respectively. The regression line is colored in red. Pearson correlation coefficient ($R$) of 0.961 (95% confidence interval: 0.954–0.968) and the $p$ value (<2.2 × 10⁻⁶, two-sided) are indicated on the plot. Source data are provided as a Source Data file.

genomic windows which were more than two standard deviations away from healthy controls in at least half of the control or cancer samples (Fig. 3d). We found a large fraction of commonly altered windows among COREAD samples for all features: 10.6% for FSI, 35.5% for MFR, and 40.2% for CNA.

To quantify the associations among the three features, we calculated their pairwise PCCs along the genome of each cancer patient. Consistent with previous findings that genomic regions with CNA exhibit more dramatic fragmentation alterations[16,22], positive correlations between FSI and CNA profiles were noted for most patients with a median PCC = 0.350 (Fig. 3e). In contrast, MFR and CNA profiles were mostly anti-correlated with a median PCC = −0.276 (Fig. 3f), probably due to global hypomethylation of the tumor genome[15]. These associations likely result from differential fractions of ctDNA in the circulating cfDNA pool shed from CNV-positive regions. FSI and MFR profiles showed weak correlations in either direction (median PCC = −0.087; Fig. 3g), which might imply complex relationships between cfDNA fragmentation and methylation that remain to be elucidated. Overall, these modest to weak associations among different feature types suggest that these data modalities can provide both concordant and complementary information, supporting the notion that integrative analysis can potentially enhance the detection power of cancer-associated signals.

## Extraction and integration of cancer-associated multimodal genomic features from cfDNA WMS for multicancer detection

To develop and validate the computational approaches for multicancer detection utilizing WMS-derived cancer genomic features, we applied WMS to plasma cfDNA samples from the MONITOR study, which comprised 780 previously untreated cancer patients and 497 healthy controls recruited from six hospitals. Visualization of healthy controls and each cancer type by t-Distributed Stochastic Neighbor Embedding (tSNE) with each feature suggested no obvious batch effects among the hospital source (Fig. S3). We randomly split the 1277 samples into a training cohort and an independent test cohort at a ratio of 7:3. The training cohort included 352 healthy controls and 542 cancer patients (46 breast (BRCA), 105 colorectal (COREAD), 42 esophageal (ESCA), 78 liver (LIHC), 110 lung (NSCLC), 83 pancreatic (PACA), and 78 gastric (STAD) cancers), 35.1% of which were at early stages (stage I or II). The test cohort consisted of 145 healthy controls and 238 cancer patients (20 BRCA, 45 COREAD, 19 ESCA, 35 LIHC, 47 NSCLC, 36 PACA, and 36 STAD), including 34.5% early-stage disease. With the entire MONITOR cohort, we obtained an overview of the pan-cancer plasma epigenome including DNA methylation (MFR) and fragmentation (FSI and FEM), as well as a pan-cancer CNA landscape (CAFF). Visualization of all four individual genomic features by tSNE plots showed clear separation of cancer patients from healthy controls, supporting their utility for cancer

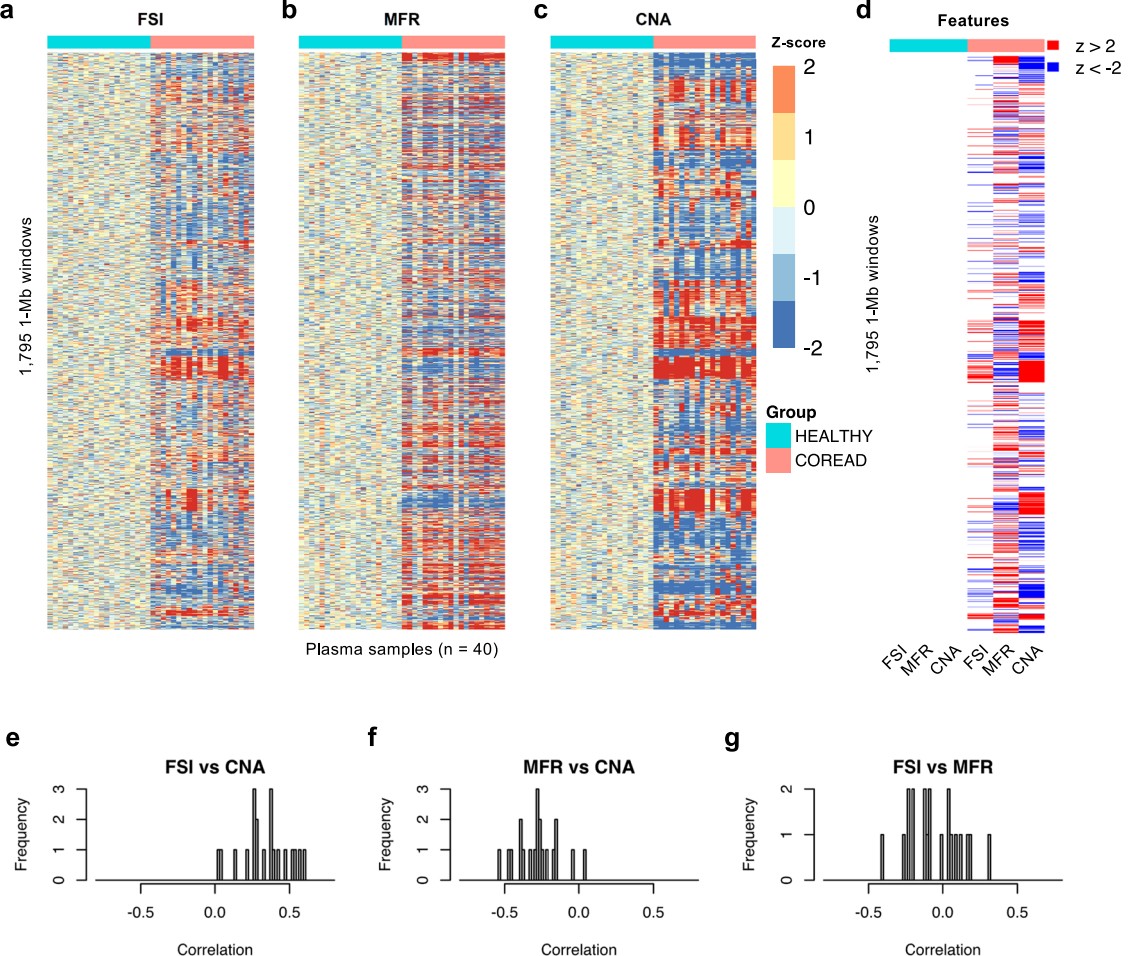

**Fig. 3 | Associations among plasma fragmentation, methylation, and CNA profiles.** A total of 20 healthy controls and 20 advanced colorectal cancer patients are analyzed. **a**–**c** Z-scores of FSI, MFR, and CNA profiles in 1795 non-overlapping 1-Mb windows across the genome (row) of healthy and cancer patient individuals (column). Genome windows are ordered by coordinates from chromosome 1 to chromosome 22. **d** For each feature type, genomic windows are indicated in red if more than half of the healthy controls or more than half of the cancer patients have z-scores above 2 and in blue if z-scores are below −2. **e**–**g** Distributions of Pearson correlations between the genome-wide FSI and CNA (**e**), MFR and CNA (**f**), and FSI and MFR (**g**) profiles for cancer patients. Source data are provided as a Source Data file.

detection. Moreover, cancer samples of the same types appeared to be separable by methylation, fragmentation size, and CNA profiles, suggesting that these features may have cancer type-specific signatures despite the arbitrary segmentation of the genome for MFR and FSI profiling (Fig. 4a). As expected, the clustering of cancer types was more conspicuous in late-stage (III and IV) than early-stage (I and II) samples, but separation of early-stage cancers from healthy controls was still discernible (Fig. S4). This result suggests the potential of these features for sensitive cancer detection across both early and advanced disease stages.

We implemented machine learning methods for MFR, FSI, and FEM to increase the sensitivity of cancer detection and all models were trained with samples from the training cohort only. To reduce the high dimensionality of feature space relative to the number of samples, principal component analysis (PCA) was performed for MFR, FSI, and FEM respectively and the top principal components (PCs) explaining most of the data variance were used as input features for model development. The performance of classification between cancer patients and healthy controls was assessed with receiver operator characteristic (ROC) analysis in the test cohort (Fig. 4b). A support vector machine (SVM) model was trained for MFR and achieved the highest classification power with an area under the curve (AUC) value of 0.947 (95% CI: 0.926–0.969) for the detection of all seven cancer

types combined, followed by FEM with a logistic regression (LR) model (AUC: 0.935, 0.911–0.958) and FSI with SVM (AUC: 0.906, 0.877–0.934). Whole-genome copy number alteration was quantified by PA score and had the lowest AUC value of 0.861 (0.825–0.897), perhaps suggesting that plasma CNA is a less sensitive feature for cancer detection. Each modality showed variable detection performances among individual cancer types as shown by ROC plots and the distribution of prediction scores (Figs. S5–S8, Panels a and b). Nevertheless, prediction scores increased with advancing cancer stage for all modalities, suggesting the association between extracted feature signal and tumor grade (Figs. S5–S8, Panel c). In addition, prediction scores of the four modalities showed moderately strong positive pairwise correlations (Fig. S9), suggesting both the concordance and complementariness among different features.

To further boost the performance of the multicancer predictive model, we constructed the ensemble THEMIS classifier with a regularized LR model to integrate all four modalities. In this model, the predictive probabilities of MFR, FSI, and FEM, along with $\log_{10}$ transformed CAFF PA scores were used as the input features, and the beta coefficients for MFR, FSI, CAFF, and FEM were 0.33, 0.34, 0.06, and 0.58 respectively, which represented the relative contribution of each modality. THEMIS outperformed individual modalities with a higher overall AUC value of 0.966 (95% CI: 0.950–0.982) in the test cohort

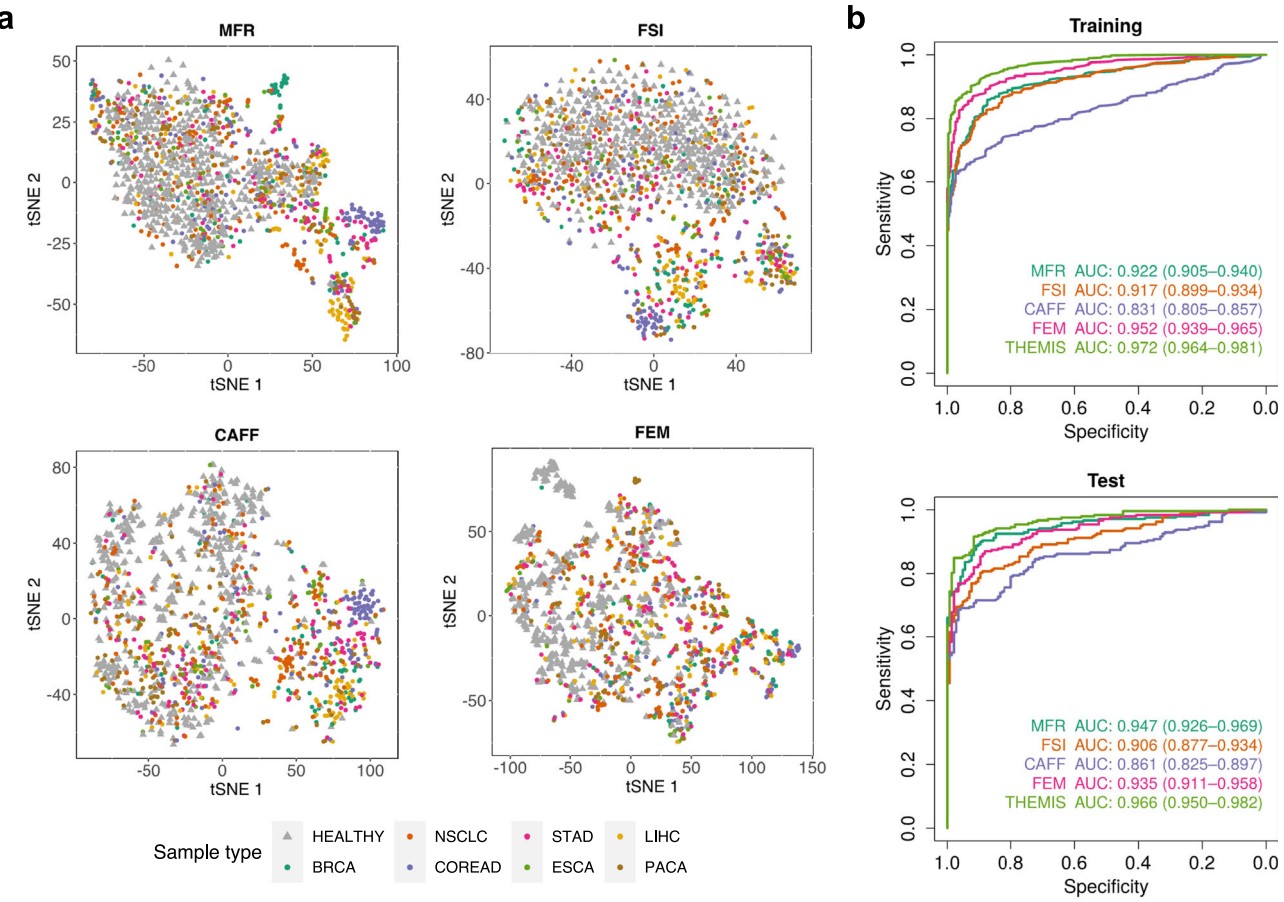

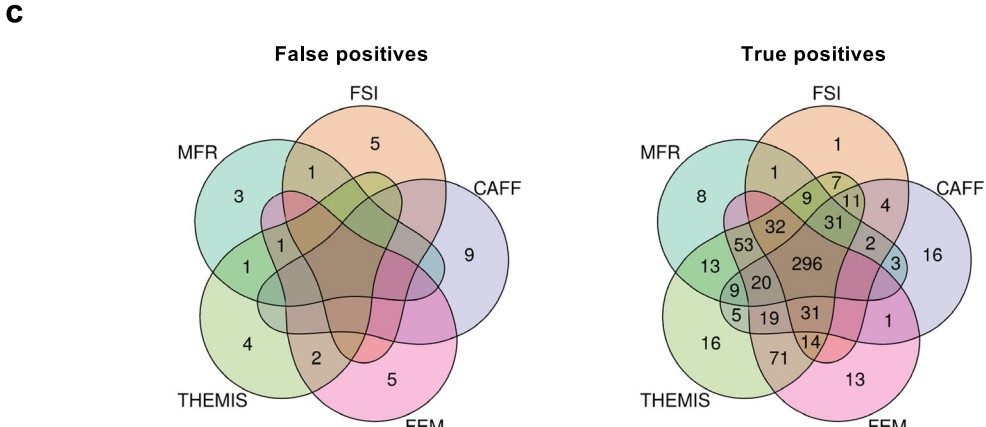

**Fig. 4 | Cancer detection by multimodal analysis of cfDNA WMS data.**
**a** Visualization of individual modalities by tSNE plots (n = 1277). Healthy control individuals and patient samples from different cancer types are annotated by shapes and colors. **b** Receiver operator characteristics for the classification of cancer patients (training n = 542; test n = 238) from healthy controls (training n = 352; test n = 145) by individual modalities and the integrative THEMIS model for the training and the test cohort respectively. The 95% confidence intervals (CIs) for AUCs are calculated with 2000 stratified bootstrap replicates. **c** Venn diagrams depicting the number of healthy individuals misclassified as cancer patients (false positives) and the number of cancer patients correctly identified (true positives) at a specificity of 99% by individual modalities and THEMIS. Samples from both the training and the test cohort are included. Source data are provided as a Source Data file.

(Fig. 4b). At a threshold of 99% specificity, THEMIS reduced the number of healthy individuals misclassified as cancer patients (false positives) and increased the number of cancer patients correctly identified (true positives) (Fig. 4c). These results confirm our hypothesis that integrative analysis of multimodal cancer genomic features can improve both the detection specificity and sensitivity over profiling individual features alone.

To investigate the minimum required sequencing depth, we analyzed 773 MONITOR samples (including 306 healthy controls and 467 cancer samples) sequenced at high depth and performed a downsampling experiment with a range from 120M to 3M reads. At each depth, we kept the original training or test group assignment for these samples and evaluated the classification performance for individual modalities and the THEMIS model following the original analysis

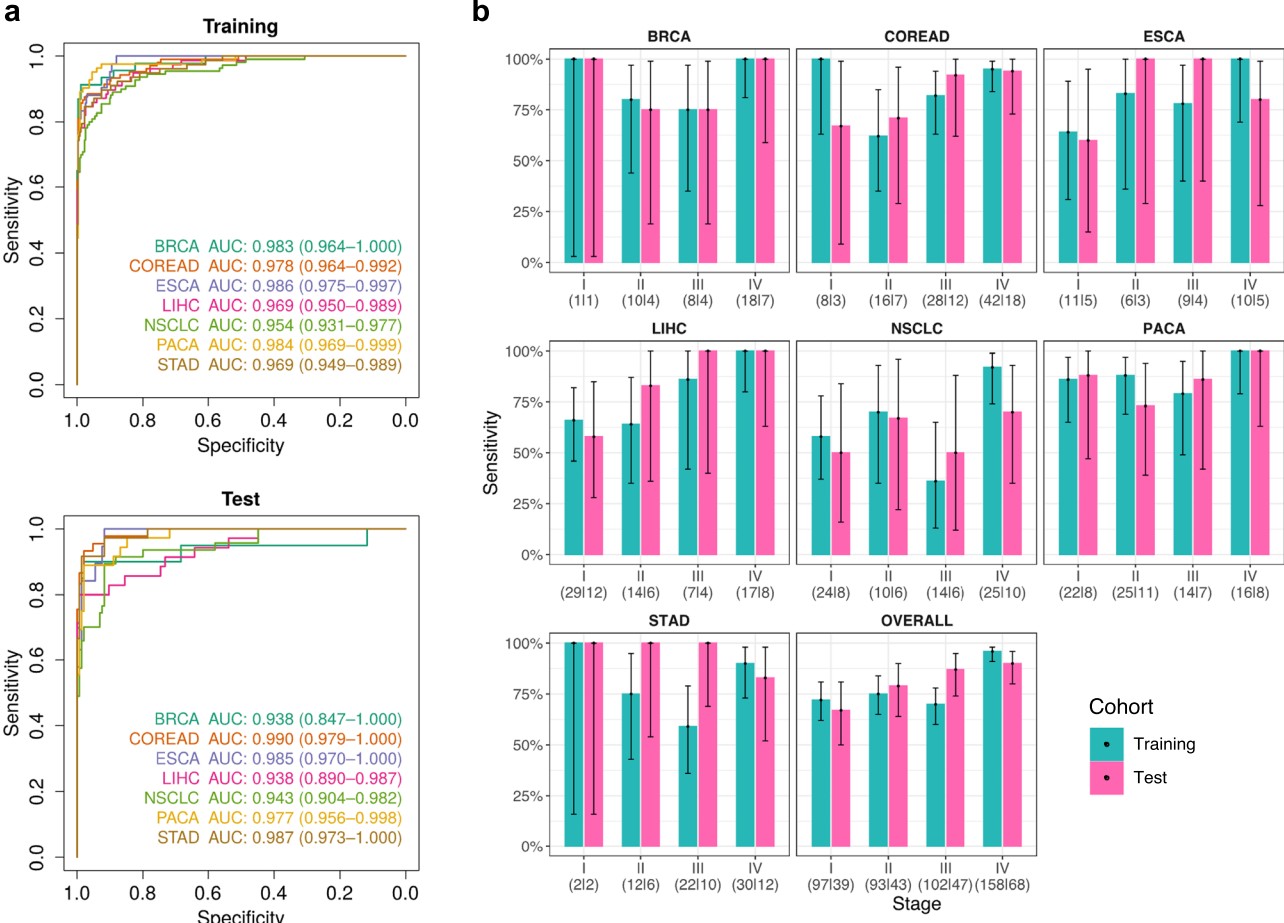

**Fig. 5 | THEMIS performance for multicancer detection. a** Receiver operator characteristics for the classification of cancer patients from healthy controls by THEMIS in the training (healthy $n = 352$; cancer $n = 542$) and the test (healthy $n = 145$; cancer $n = 238$) cohort split by cancer type. The 95% confidence intervals (CIs) for AUCs are calculated with 2000 stratified bootstrap replicates. **b** Detection sensitivity of individual cancer types by clinical stage. Sensitivity at a specificity of 99% is depicted with bar charts. Error bars represent 95% Wilson confidence intervals. The numbers of samples in the training and the test cohort for each stage are indicated below the plots and separated by a vertical line. Cancer samples with unknown stages are omitted from display. Source data are provided as a Source Data file.

pipeline. For MFR, FSI, and FEM, we found an inverse relationship between sequencing depth and the number of PCs to explain data variance (Fig. S10a), suggesting that increasing sequencing depth may improve the signal-to-noise ratio for feature extraction. Accordingly, overfitting of both individual and integrative models was notable at lower depth (Fig. S10b). With 60M reads, model performance reached plateau without obvious overfitting, indicating that 60M reads were required by THEMIS for reliable feature extraction and model development.

### Performance of THEMIS for multicancer early detection

THEMIS achieved high AUC values in the detection of all seven types of cancer ranging from 0.938 for BRCA (95% CI: 0.847–1.000) and LIHC (95% CI: 0.890–0.987) to 0.990 (95% CI: 0.979–1.000) for COREAD in the test cohort (Fig. 5a). Remarkably, THEMIS maintained consistent AUC values between the training and the test cohort for every cancer type, suggesting minimal overfitting of the ensemble classifier. At a specificity of 99%, THEMIS correctly classified 440/542 (81% sensitivity; 95% CI: 78–84%) cancer patients in the training cohort versus 197/238 (83% sensitivity; 95% CI: 77–87%) in the test cohort (Figs. 5b and S11a). As with individual modalities, the prediction score and detection sensitivity of THEMIS increased with increasing clinical stage (Figs. 5b and S11b), implying that cancer-like signals profiled by THEMIS closely correlated with tumor

load. Particularly, the overall detection sensitivity of THEMIS for early-stage (I and II) cancers was 74% (95% CI: 67–79%) and 73% (95% CI: 63–82%) in training and test, respectively.

To investigate the clinical sensitivity of our approach more precisely, we adopted the concept of clinical limit of detection (cLOD)[24] to evaluate classifier performance. We used a customized 769-gene panel to calculate the variant allel frequency (VAF) of 65 plasma samples from the MONITOR cohort with available paired tumor tissue and white blood cell (WBC) samples, and single nucleotide variant (SNV) was detected in both tissue and cfDNA after removal of WBC background. We defined the clinical LOD as the mean VAF at which the probability of positive cancer signal detection was at least 50% with a 99% specificity. The cLOD of the THEMIS classifier ($1.5 \times 10^{-3}$ mean VAF, 95% CI: $2.8 \times 10^{-4}$–$3.7 \times 10^{-3}$) was lower than individual modalities (Fig. S12), again supporting our hypothesis that feature integration can boost detection sensitivity.

To evaluate the robustness of the classifiers to potential batch effects for prospective clinical application, we investigated the impact of sample source on model performance by selecting training and testing samples collected from different sites. Specifically, all cancer (except BRCA) and healthy cohorts in our data were recruited from two to four hospitals (Supplementary Data 4) and therefore, for each non-BRCA cancer or healthy cohort, samples from one hospital were used for model testing and samples from the

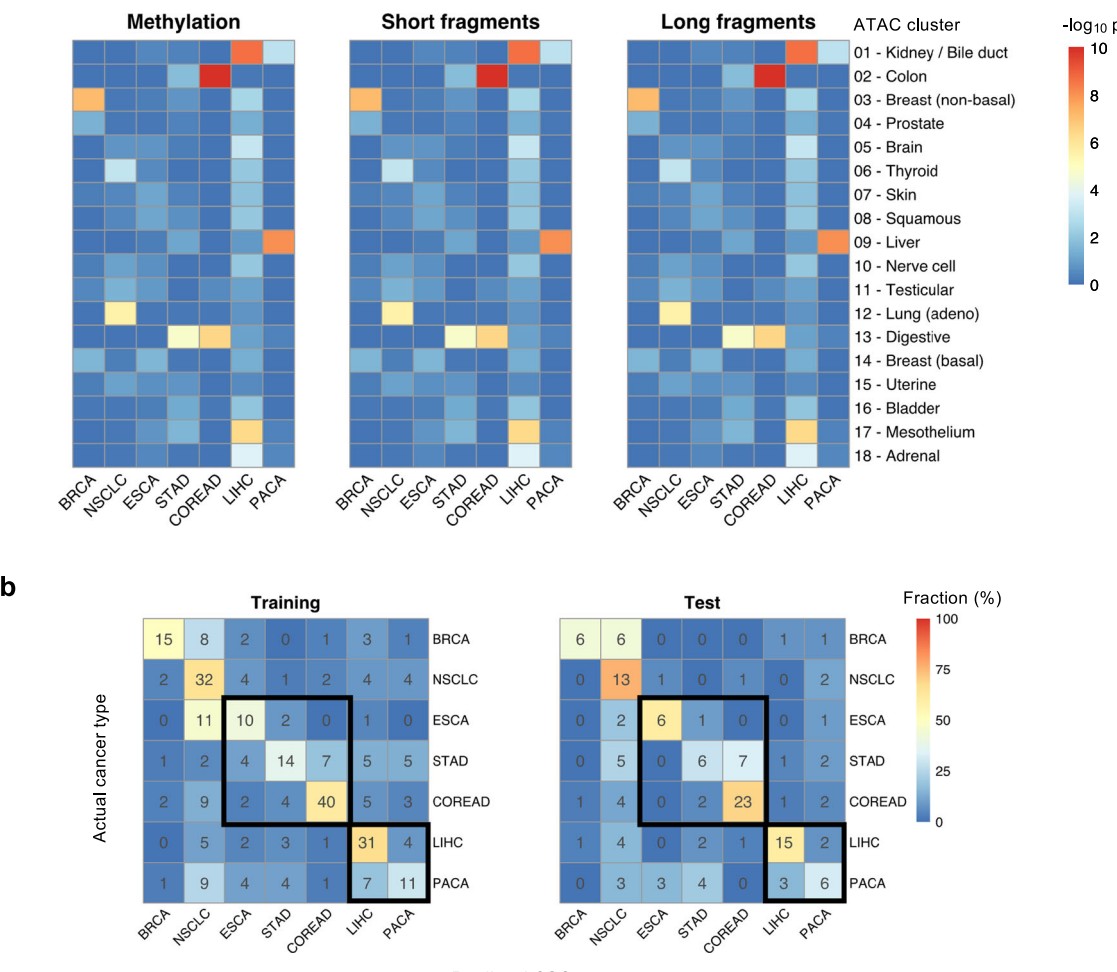

**Fig. 6 | Classification of cancer signal origin. a** Heat maps of *p* values (one-sided Wilcoxon test) of methylation levels, short fragment coverage, and long fragment coverage of each ATAC-seq cluster between individual cancer types and other cancers in the training cohort. Each feature is normalized as described in "Methods". Training cancer samples correctly identified as true positives by THEMIS score at 100% specificity are used for analysis, including 30 BRCA, 49 NSCLC, 24 ESCA, 38 STAD, 65 COREAD, 46 LIHC, and 37 PACA samples. **b** Confusion matrices representing the accuracy of CSO localization in the training and the test cohort. Color corresponds to the proportion of predicted CSO calls. Test cancer samples correctly identified as true positives by THEMIS score at 100% specificity are used for evaluation of the CSO classifier, including 14 BRCA, 17 NSCLC, 10 ESCA, 21 STAD, 33 COREAD, 25 LIHC, and 19 PACA samples. Closely associated cancer types are enclosed within squares. Source data are provided as a Source Data file.

remaining hospitals for model training. This scheme yielded a total of 384 split-by-hospital combinations, and we implemented feature extraction and model training for each combination. We observed comparable AUCs among the 384 combinations for individual modalities and the integrative THEMIS model (Fig. S13a), suggesting little impact on classifier performance by sample collection site. Instead, we found that model performance appeared to be inversely correlated with the proportion of early-stage (I or II) cancer patients in each combination (Fig. S13b), consistent with the notion that cancer detection power is dependent on ctDNA fraction in the blood.

**Methylation and fragmentation profiles of accessible chromatin regions inform the tissue origin of cancer signals**
With static genome sequence, the epigenome is cell type and tissue specific and may inform the origin of circulating cfDNA[11,12,16]. The utility of cfDNA methylation in tumor localization has been recently shown by targeted sequencing[11] or immunoprecipitation enrichment[12]. Unlike these methods, analysis of single CpGs is beyond the scope of shallow WMS data despite its base-pair resolution. Recently, the chromatin

accessibility landscape of 23 primary cancer types was profiled by ATAC-seq assay using TCGA samples[23]. Many of the ATAC-seq peaks overlap TSS-distal regulatory elements such as enhancers, which play critical roles in oncogenesis and cancer type specificity[25,26]. Therefore, these ATAC peaks were grouped into 18 clusters by cancer tissue specificity[23], which allowed us to profile the cfDNA methylation and fragmentation patterns within accessible *cis*-regulatory elements for localization of cancer signal origin (CSO). To develop the multicancer CSO classifier, we used cancer samples detected by THEMIS at 100% specificity, which included 289 training and 139 test samples. For each sample, we characterized the mean cfDNA methylation and the aggregate coverage of short (100–166 bp) and long (169–240 bp) fragments within all ATAC-seq peaks of each cluster. We observed lower methylation of ATAC-seq clusters in relevant cancer types compared with other cancers (Fig. 6a), which is consistent with published reports on the anticorrelation between DNA methylation and chromatin accessibility[23,27] and provides orthogonal validation of the tissue specificity annotated for ATAC-seq clusters. For example, COREAD samples were notably hypomethylated in Cluster 2 (Colon) and both COREAD and STAD samples were hypomethylated in Cluster

13 (Digestive). Likewise, BRCA and NSCLC samples had lower methylation in Cluster 3 (Non-basal breast) and Cluster 12 (Lung), respectively. We also observed lower coverage of short and long fragments in cancer relevant clusters (Fig. 6a), suggesting increased chromatin accessibility in these regions. Based on these three features, the seven cancer types investigated by this study were classifiable using a random forest algorithm. This model correctly localized 153/289 (53%, 95% CI: 47–59%) cancer samples in the training cohort and 75/139 (54%, 95% CI: 46–62%) cancer samples in the test cohort (Fig. 6b). Not surprisingly, some of the misclassified samples were assigned into highly associated tissues. For example, gastric cancers were frequently misclassified as colorectal cancer. We therefore combined closely associated cancer types for evaluating the accuracy of CSO assignment, including merging ESCA, STAD, and COREAD into a digestive tract cancer group and merging LIHC and PACA into a hepatopancreatic cancer group. After this grouping, our CSO classifier achieved an accuracy of 183/289 (63%, 95% CI: 58–69%) for the training cohort and 90/139 (65%, 95% CI: 57–72%) for the test cohort.

## Discussion

In this study, we developed a multicancer detection and localization approach, THEMIS, by leveraging crucial tumor genetic and epigenetic characteristics with shallow plasma WMS. We showed proof that the mild enzyme-based WMS platform is applicable for cfDNA fragmentation and coverage analyses, which enables simultaneous profiling of multiple types of cancer genomic features such as methylation, fragmentation, and CNA. We integrated these modalities and built an ensemble classifier for cancer detection, which achieved high sensitivity across disease types and stages. We also profiled plasma methylation and fragmentation patterns at cancer tissue specific accessible chromatin regions, which allowed us to accurately localize the origin of cancer signals. Together, these results demonstrate the promising clinical utility of THEMIS in multicancer early detection and localization.

Several blood-based multicancer detection approaches have been recently published, including targeted methylation sequencing[11], immunoprecipitation-based methylation enrichment[12], genome-wide fragmentation profiling[16], and detection of mutation and serum protein biomarkers[28]. The advent of enzymatic reaction based methylation sequencing allows for simultaneous analysis of additional cfDNA epigenetic features besides methylation, as validated for nucleosome footprint analysis by cfNOMe[29] and for fragmentation and CNA profiling by this study. While both cfNOMe and THEMIS employed the same enzymatic methylation sequencing assay, cfNOMe profiled a small cohort of cfDNA samples from renal disease patients with high fraction of circulating kidney-derived DNA (>1%). By contrast, THEMIS was developed with a large clinical cohort of multicancer patients wherein the fraction of circulating tumor DNA can be lower than 0.1% in early-stage patients. Therefore, the application of cfNOMe for cancer detection remains to be investigated. In addition, compared with THEMIS, much higher sequencing depth was needed by cfNOMe to reliably analyze nucleosome footprinting, which may limit its cost-effectiveness for clinical application. Recently, the concept of integrating multiple types of cancer features for more sensitive ctDNA detection has also been explored by another bisulfite-free whole-genome methylation sequencing method cfTAPS[30]. Unlike cfTAPS which employed a lengthy incubation step with pyridine borane to convert 5caC to DHU, the enzyme-based WMS assay is anticipated to be more robust because it is potentially less sensitive to fluctuations in experimental conditions. Also, in contrast to the deep sequencing of cfTAPS for methylome analysis[30], THEMIS model was developed on low-pass sequencing with only 60 million paired reads (~2×genome coverage) to control sequencing cost, and the performance and generalization capacity of both individual modalities and the integrative THEMIS model were thoroughly evaluated with a large clinical cohort comprising seven major cancer types across all stages. Together, these

technical advances over similar studies highlight the potential of THEMIS for real-world clinical application.

A blood-based multicancer detection test presents several major advantages over currently available imaging-based diagnostic procedures, such as the ability to detect small and early tumors, the reduction of accumulative false positive rates for screening multiple organs, and improved patient compliance with its minimal invasiveness. Therefore, THEMIS is intended for a general screening population with low to average cancer risk, which requires a sufficiently high specificity to minimize the false positive rate (FPR). At a fixed training specificity of 99%, THEMIS had an overall sensitivity of 83% in the test cohort. When applied to a population with a 1.5% cancer incidence rate per year (which approximates the cancer incidence rate for people over 50 years old in the United States in 2019[31]), THEMIS would theoretically detect 1245 true positives and result in 985 false positives per 100,000 participants, yielding a positive predictive value (PPV) of 56%. Although a prospective screening study is needed to precisely investigate the PPV of our approach, this preliminary PPV calculation demonstrates the potential of THEMIS for population-level cancer screening with minimized risks of false positives.

The concentration of ctDNA is generally low especially in early-stage disease. To more sensitively capture weak tumor signals, machine learning methods are frequently implemented in published studies which are prone to overfitting issues arising from technical and biological noises. Although additional functional validations may be needed to further prove the fidelity of feature extraction and modeling by THEMIS, multiple lines of evidence suggest that THEMIS characterizes bona fide tumor-derived signals. First, because alteration of genome copy number is a highly cancer-specific event[17,32,33], CNA may be utilized to confirm the fidelity of cancer signals we detected. Indeed, we observed strong correlations between fragmentation and copy number in COREAD samples, consistent with other studies[16,22,34]. We also noted anticorrelations between methylation and copy number, consistent with the global hypomethylation of tumor DNA[15]. Second, we noticed consistent cancer detection performance between the training and the test cohort with both the base models of individual modalities and the integrative THEMIS model. To estimate variability of the classifiers, we performed 100 random splits of the MONITOR cohort into training and test cohorts at the fixed ratio of 7:3 and implemented feature extraction and model development for each split. Model AUCs were highly consistent between the training and the test cohorts per run as well as over the 100 runs for all classifiers (Fig. S14), demonstrating the stability of our approach. Finally, prediction scores and detection sensitivity of THEMIS increased with increasing disease stage. These results imply that signals detected by THEMIS likely originates from tumor cells instead of background noise.

Localizing the origin of cancer signal is a crucial component of a multicancer test to guide follow-up diagnostic workups. Tissue specificity has been reported for multiple plasma epigenetic features including methylation, fragmentation, and nucleosome footprint[11,16,35], among which deep targeted methylation sequencing showed the most robust CSO classification performance[11]. Considering the low depth of our WMS approach, THEMIS adopted an alternative strategy by employing epigenetic features of the large number of tissue specific accessible chromatin regions, most of which are putative noncoding regulatory elements such as enhancers[23]. As expected for open chromatin, we observed reduced DNA methylation and fragment coverage in most cancer type-relevant clusters (e.g., Cluster 2 for COREAD) in plasma, which suggests the concordance between tumor tissue and plasma epigenome. A notable exception is Cluster 9 (Liver), with which we only observed weakly reduced methylation in our LIHC samples. This might result from heterogeneity in the etiology of liver cancer between the Western and Chinese populations analyzed in the two studies, which are primarily driven by alcohol consumption and HBV infection, respectively. The coverage of short and long fragments for

Cluster 9, however, was significantly reduced in our LIHC samples. This discrepancy between DNA methylation and fragmentation patterns requires further investigation; nevertheless, it demonstrates the utility of multimodal epigenome profiling for more comprehensive understanding of the regulatory mechanism underlying tumorigenesis as well as for more sensitive ctDNA detection. One of the main limits to our current CSO performance is probably the diversity of types, subtypes, and etiologies of available cancer chromatin accessibility datasets and therefore, this study mainly presented a proof of concept for the feasibility of CSO classification by shallow WMS using integrative epigenetic profiling of regulatory elements. In addition, although accessible regulatory elements generally tend to be hypomethylated, the clustering of ATAC-seq peaks by methylation profiles may be further refined with cancer tissue methylation datasets. We anticipate the availability of chromatin accessibility as well as methylation datasets from more diverse cancer types, subtypes, and etiologies to facilitate the refinement of our CSO marker selection and the improvement of prediction accuracy.

Despite the proof of concept for a sensitive multicancer detection and localization approach, this study has several limitations. Due to the small sample size, participant demographics were not completely matched between cancer patients and noncancer controls. Also, although our approach displayed high sensitivity for early-stage cancers, further assessment is needed considering the relatively limited size of early-stage samples in the cohort. In addition, we lack complete one-year follow-up information for participants considered healthy at the time of writing this manuscript, therefore we cannot rule out the possibility that some control individuals bear early-stage cancer thus overestimating the false positive rates of the models. Because this is a retrospective case-control study, precise assessment of the real-world performance of THEMIS (including sensitivity, specificity, PPV, etc.) and the establishment of its clinical utility require future investigations in a larger prospective screening cohort with complete long-term follow-up.

## Methods

The ethics committee of National Cancer Center approved the protocol of this study (NCC–007821), and our research complies with all relevant ethical regulations. All participants gave their written informed consent for research use.

### Study design and participants

Plasma samples of the MONITOR cohort were prospectively collected from 497 healthy donors and 780 patients with breast, colorectal, esophageal, gastric, liver, lung, or pancreatic cancers of various stages and analyzed retrospectively. Participants were enrolled from six hospitals including the Cancer Hospital Chinese Academy of Medical Sciences. Participant sex was not considered in the study design. Individuals were considered healthy if they had no previous history of cancer and negative clinical diagnosis at the time of administration. Plasma samples from cancer patients were obtained before tumor resection or therapy. Participants of the MONITOR cohort were randomly assigned into a training cohort and a test cohort at a ratio of 7:3. Clinical information of participants including age, gender, and disease stage were summarized in Supplementary Data 1. Patients with unknown disease stage were indicated as stage NA.

### Isolation of plasma cfDNA

Plasma samples were isolated from 10 ml of peripheral blood collected and stored in Cell-Free DNA Storage Tube (Cwbiotech). Blood was centrifuged at $1600 \times g$ for 10 min at 4 °C and plasma was transferred to new 1.5 ml tubes (AXYGEN). A second centrifuge was performed at $16,000 \times g$ for 15 min at 4 °C to remove remaining cell debris and a total of ~4 ml of plasma was obtained and stored at −80 °C until use. cfDNA was extracted using MagMAX Cell-Free DNA Isolation Kit (Thermo

Fisher Scientific, Catalog# A29319) per manufacturer instructions. The quantity and quality of extracted cfDNA was assessed with Bioanalyzer 2100 (Agilent).

### Whole-methylome sequencing of plasma cfDNA

The entire amount of extracted plasma cfDNA (capped at 30 ng if more was extracted) was used to generate WMS libraries with NEBNext Enzymatic Methyl-seq Kit (New England Biolabs, Catalog# E7120) per manufacturer instructions with the modification that 100 ng of carrier RNA (TIANGEN, Catalog# CA-310/PA-310) was added before denaturation by sodium hydroxide. Libraries were amplified with 9 cycles of PCR reactions and quantified using Qubit dsDNA HS Assay Kit (Thermo Fisher Scientific). Libraries were sequenced on NovaSeq 6000 (Illumina) with read length of paired-end 100 bp.

### Methylation sequencing data processing

Methylation sequencing reads were demultiplexed by Illumina bcl2fastq (v2.20.0) and adapters were trimmed by Trimmomatic (v0.36). Reads were aligned against the human reference genome (hg19) and deduplicated by BisMark (v0.19.0)[36]. Samtools (v1.3) and BamUtil (v1.0.14) (https://github.com/statgen/bamUtil) were used for sorting and overlap-clipping of mapped reads. Reads with mapping quality below 20 were filtered out. To normalize for sequencing depth among samples, 60 million paired reads were randomly selected from each sample and used for downstream analyses.

### Copy number profiling

Hg19 autosomes were divided into adjacent, nonoverlapping 100-kb bins and the coverage of each bin was calculated. A LOESS-based method[17] was applied to correct for coverage bias related with GC content of the reference genome. Bins overlapping the Duke blacklisted regions (http://hgdownload.cse.ucsc.edu/goldenpath/hg19/encodeDCC/wgEncodeMapability/) or the hg19 gap track (downloaded from UCSC Table Browser) were prone to low mapping quality and removed from analysis. Bins with original coverage over 100 but zero coverage after GC-correction were also excluded. To remove bins with copy number variations among healthy controls, we calculated the z-score of each GC-corrected bin over the 352 healthy controls in the training cohort (i.e., baseline samples). Bins with absolute z-scores above two among more than 60 baseline samples or above four among any baseline samples were filtered out.

### Fragment size index (FSI)

To profile differences in cfDNA fragmentation size between cancer patients and healthy controls, the frequency of each fragment size was calculated for individual samples and Wilcoxon one-sided test was performed between cancer samples and healthy controls in the training cohort to select for differentially represented fragment sizes (Fig. S15). At a $p$ value threshold of 0.01, most fragment sizes within 50–300 bp exhibit frequency differences except 167–168 bp and 241–259 bp. Therefore, fragments within 100–166 bp were considered as short fragments and fragments within 169–240 bp were considered as long fragments. The ratio of short to long fragment counts was defined as fragment size index (FSI).

To characterize the genome-wide FSI profile, hg19 autosomes were divided into 100-kb A/B compartments representing open and closed chromatin regions[37] and those overlapping the Duke blacklisted regions or the hg19 gap track were removed from analysis. Counts of short and long fragments within each bin was calculated and the ratios of short to long fragment counts were corrected against GC content of the reference genome using a LOESS-based method[17]. After merging adjacent 100-kb bins, genome was segmented into 502 nonoverlapping 5-Mb windows (Supplementary Data 5) and the FSI of each 5-Mb window was defined as the average of its overlapping GC-corrected 100-kb FSI values.

### Methylated fragment ratio (MFR)

Hg19 autosomes were tiled into 1846 adjacent, nonoverlapping 1-Mb windows after filtering out genomic regions overlapping the Duke blacklisted regions or the hg19 gap track (Supplementary Data 6). Read pairs were merged into fragments and those failing to meet the following criteria were discarded: (1) Fragment covers at least 3 CpGs. (2) Fragment length is between 80 and 250 bp, which is the size range of most cfDNA molecules. (3) Conversion rate of non-CpG cytosines exceeds 95%. The methylation level of each 1-Mb window was quantified by the fraction of fragments with fully methylated CpGs, or methylated fragment ratio (MFR).

### Associations among fragmentation, methylation, and copy number profiles

The 1846 1-Mb windows determined by MFR analysis were used to investigate the associations among methylation, fragmentation, and CNA across the genome. The FSI of each window was calculated as the average GC-corrected FSI values of its overlapping 100-kb bins. For CNA analysis, the coverage of each window was calculated as the total GC-corrected coverage of its overlapping 100-kb bins. Because of the more stringent filtering criteria for CNA analysis, 1795 windows were eventually kept for association studies. Each feature type for a sample was quantile-normalized with that of healthy controls. A z-score was calculated for each bin over the corresponding bins of healthy controls.

### Chromosomal aneuploidy of featured fragments (CAFF)

CAFF analysis followed the same procedures as CNA profiling except that fragments shorter than 151 bp or longer than 220 bp were extracted for coverage analysis to increase the presence of tumor-derived abnormal fragments[19,20]. The coverage of each chromosome arm was calculated by summing the GC-corrected coverage of all its 100-kb bins. To summarize chromosome arm-level copy number changes, we adopted a previously described approach to calculate the plasma aneuploidy (PA) score of each sample using the five chromosome arms exhibiting the most dramatic copy number alterations from baseline samples[17].

### Fragment end motif (FEM)

The 4-nucleotide (i.e., 4-mer) fragment 5′ end motif was extracted as previously reported[18] with the following modifications: (1) Fragments shorter than 171 bp were selected for analysis. (2) Only reads mapped to the Crick strand were used for calculation.

### Visualization by tSNE

MFR, FSI, and CAFF values of individual samples were transformed to z-scores and the FEM frequencies of each sample were quantile-normalized with the healthy controls. To reduce the dimensionality of data for visualization, t-distributed stochastic neighbor embedding (tSNE) was applied for individual modalities with the Rtsne (v0.16) package for R (v4.0.2) with the following parameters: dims = 2, pca = F, max_iter = 2000, theta = 0.05, perplexity = 10.

### Machine learning models for MFR, FSI, and FEM

An in-house Python (v3.7.3) script based on sklearn module (v1.2.0) was used for the development of machine learning models. To mitigate overfitting, 10 bootstraps of the training cohort were performed for hyperparameter optimization wherein each bootstrap randomly selected 70% of the training cohort for model training and the remaining 30% for model validation. The average probability of the resulting 10 sub-models was used as the final prediction score of a sample. Model performance for each modality was evaluated with the independent test cohort.

**MFR**. To construct an MFR classifier, principal component analysis (PCA) was performed for the MFR values of the 1846 windows to reduce feature dimensionality within each bootstrap. For each sub-model, top principal components (PCs) explaining at least 95% of the variance of the training samples were used for model development. A support vector machine (SVM) model was trained with the training samples under tenfold cross-validation and validated with the validation samples.

**FSI**. For each sample, the FSI values of the 502 5-Mb windows were normalized to z-scores across the genome. PCA and the training of an SVM model with top PCs explaining at least 90% of data variance followed the procedures of MFR model development.

**FEM**. PCA of the 256 4-mer end motif frequencies and the training of a logistic regression (LR) model with top PCs explaining at least 95% of data variance were performed following the procedures of MFR model development.

For each modality, the number of top PCs employed for model training was selected by determining the optimal data variance of the training folds explainable by the PCs, which could yield high validation AUCs with minimal overfitting across the 10 bootstraps (Fig. S16).

Considering the relatively low feature dimensionality and the moderate sample size, SVM, LR, and random forest (RF) models were tested for MFR, FSI, and FEM modalities and similar performances were observed. The model with the best performance was selected for each modality.

### Development of THEMIS classifier for feature integration

To integrate the predicted cancer probability scores by MFR, FSI, and FEM along with $\log_{10}$ transformed PA scores of CAFF, a generalized linear model (GLM) with elastic-net penalization was constructed using the R package CARET under 20-fold cross-validation within the training cohort for hyperparameter tuning. Given the small size of input features, we chose to apply the parametric statistical model LR to better interpret the contribution from each modality. The resulting THEMIS model calculates the probability of cancer in a participant as:

$$\Pr(cancer) = \exp(Z)/(1 + \exp(Z))$$

where $Z = 0.57 + 0.33 * MFR + 0.34 * FSI + 0.06 * CAFF + 0.58 * FEM$

Prediction scores by individual modalities and the integrative THEMIS classifier were listed in Supplementary Data 4 for the MONITOR cohort.

### Clinical limit of detection

Targeted sequencing with a customized panel of 769 cancer-related genes was performed for somatic mutation identification[38]. Briefly, between 30 and 300 ng of fragmented tumor tissue or WBC genomic DNA or between 10 and 50 ng of cfDNA was used for library construction with KAPA Hyper Prep Kit (Roche, Catalog# KK8504). Targeted regions were captured with HyperCap Target Enrichment Kit (Roche) according to manufacturer instructions. The enriched libraries were amplified with KAPA HiFi HotStart ReadyMix (Roche, Catalog# KK2602) and sequenced on Illumina Novaseq 6000 in 150-bp paired-end mode.

To determine the detection sensitivity of a classifier at 99% specificity, a logistic regression was estimated between classifier predictions versus $\log_{10}$(mean VAF) for 65 cancer plasma samples with corresponding tumor tissue and WBC samples. Clinical LOD was defined as the mean VAF for a cancer detection probability of 50%. The 95% confidence interval was estimated using a Gaussian approximation for the standard error of the logistic regression slope.

### Cancer signal origin classification

We chose cancer samples identified as true positives by THEMIS at 100% specificity to develop and evaluate the CSO classifier. Plasma

methylation and fragmentation profiles for 18 clusters of tissue-specific 500-bp open chromatin peaks identified by ATAC-seq[23] were used for model development.

For each sample, methylation level was quantified by the fraction of methylated CpGs within individual ATAC-seq peaks. Because open chromatin regions tend to be hypomethylated[23,27], Wilcoxon one-sided test was performed between cancer samples and healthy controls in the training cohort to select for peaks with lower methylation among cancer patients ($p < 0.05$), which constituted ~90% of the original ATAC-seq peaks, for CSO analysis. For each cluster, the average methylation level of the remaining peaks was computed and transformed into a z-score over the corresponding cluster of healthy controls in the training cohort. Subsequently, the z-scores of 18 clusters were scaled between 0 and 1 for each sample.

To characterize the fragmentation profile of each sample, the aggregate counts of short (100–166) and long (169–240 bp) fragments falling within the peak regions of each cluster were calculated respectively and transformed into z-scores over the corresponding cluster of healthy controls in the training cohort. The z-scores of 18 clusters were then scaled between 0 and 1 for each sample.

Finally, the normalized methylation level, short fragment coverage, and long fragment coverage of each cluster were used as the input features for the development of the multi-class CSO classifier. Because the feature dimensionality was relatively high compared with the sample size, to avoid overfitting an RF model was trained using RandomForest package in R with 2000 trees under leave-one-out cross-validation.

### Statistics and reproducibility
All statistical analyses were performed using R version 4.0.2. To estimate the required sample size, power calculations suggest that an analysis of approximately 800 patients with cancer and 500 healthy controls suffices for estimation of a sensitivity of 0.8 with a margin of error of 0.05 at a specificity of 0.95. No data were excluded from the analyses. The experiments were not randomized, and the investigators were not blinded to allocation during experiments and outcome assessment.

### Reporting summary
Further information on research design is available in the Nature Portfolio Reporting Summary linked to this article.

## Data availability
The 1277 cfDNA WMS data of the MONITOR cohort generated in this study have been deposited in the Genome Sequence Archive (GSA) for Human database under accession code HRA003209. To protect patient privacy, data access can be obtained through a request to the data access committee. Access to the data will be restricted to non-commercial entities. Access will be provided within approximately one week and be available for one year. Tissue ATAC-seq peaks were download from TCGA [https://gdc.cancer.gov/about-data/publications/ATACseq-AWG]. Reference genome hg19 was used for mapping samples. Source data are provided with this paper.

## Code availability
Processed data and code to reproduce the figures are publicly available and deposited in GitHub [https://github.com/yulongbio/themis.git].

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

## Acknowledgements

The authors are grateful to all patients and their families for their voluntary participation in this study. This work was supported by the National Key R&D Program of China (2021YFC2500900, S.G.), CAMS Initiative for Innovative Medicine (2021-I2M-1-015, S.G. and F.T.), Central Health Research Key Projects (2022ZD17, S.G.), CAMS Innovation Fund for Medical Sciences (2021-I2M-1-061, F.T.), National Natural Science Foundation of China (81871885, F.T.), and National Key R&D Program of China (2021YFC2500400, W.C.).

## Author contributions

F.B., Z.W., Yulong Li, and W.G. designed the study and wrote the main manuscript text. Y.H. oversaw next-generation sequencing experiments. T.H., F.L., S.Y., S.L., X.L., and P.N. performed data analysis and figure preparation. S.X., R.Z., M.Z., P.S., F.F., W.G., J.D., G.B., Yuan Li, Q.H., and B.Z. enrolled patients and collected clinical data. Y.S.H., W.C., F.T., S.G. oversaw the study. All authors reviewed and discussed the manuscript.

## Competing interests

Yulong Li, Y.H., T.H., F.L., S.Y., P.N., Y.S.H. and W.C. are employees of Genecast Biotechnology Co., Ltd. These authors have filed a patent application (PCT/CN2022/098450) related to the methods described in this manuscript, which is granted in China (CN114045345B: Detection system and detection method of genomic carcinogenesis information based on cell-free DNA) and is pending in other countries. All other authors have declared no competing interests.
