## [Peer Review File · Nature Communications]

REVIEWER COMMENTS

Reviewer #1 (Remarks to the Author): expert in DNA fragmentation analysis

The author utilized multi-modality information from the enzymatic conversion based low-coverage whole-methylome sequencing for the multi-cancer early detection. Overall, it is an interesting study. However, I have some major concerns of the study:

1. Downsample experiment, what is the minimum required sequencing depth for each modality and what is that for the integrate multi-modality? Maybe a downsampling experiment is needed.
2. Need to share the raw data and fragment-level data. There is no way that other people can replicate their study. "available upon request" usually means data is not available to the community.
3. In the multi-modality classification, Which modality contributes most? Need to show the fraction of power from each modality in the ensemble model.
4. When assessing the concordance between WGS and WMS, do they have similar sequencing depth? Only one pair of sample is shown, not sure how stable it is, the author needs the replicates.
5. The title is about multi-cancer early detection. However, only ~30% of samples are early-stage. This is misleading. For Figure 4A, the author need to separate the stage and show it again. I suspect the separation is mainly just in late-stage. Otherwise, the description might be misleading to the reader that early-stage signals here are also clearly distinguished between cancer types.
6. The best tuned elastic-net mixing parameter alpha. Did they tune the parameter based on the test cohort or cross validation within the training cohort? This is not clearly described.

Minor:

1. How is the bisulfite conversion rate of the WMS data?
2. 4 ml of plasma is a lot. Don't have to emphasis on it in the abstract...

Reviewer #2 (Remarks to the Author): expertise in machine learning methods

THEMIS: a multicancer early detection platform integrating multimodal information from cell-free DNA whole-methylome sequencing

Bie et al present a methylation + machine learning method called THEMIS, which enables cancer detection from multimodal cfDNA signals. Their study includes application of enzymatic methyl-seq (EM-seq), which was recently published as an alternative to bisulfite sequencing, to a total of 1277 plasma cfDNA samples at ~2x coverage. These data are subjected to various machine learning analyses, integrating methylation, fragmentation, copy number and fragment-end motif information, to show very good performance separating cancer from healthy control samples and feasibility to detect the origin of the cancer. The work has a lot of merit and I consider it a worthy addition to the rapidly developing field. Some concerns need to be address / clarified in order to make it suitable for publication:

- Data and code availability - the authors should better adhere to the data availability requirements of the journal (and of modern research to be honest). The manuscript would be much more valuable if data are deposited in a (controlled access) repository with clear guidelines on the criteria for access (and not only available upon reasonable request). Same holds for the code and scripts (machine learning, sequencing data processing, etc), for which there is no mention on how it is available for the purpose of reproducing the results in the manuscript and for utilisation by the scientific community. I read that there is a patent filed based on this work. That is absolutely fine, but this should not (and does not have to) conflict with non-commercial (academic) reproducibility and reuse of this work.

- I think there should be more discussion around tumor fraction. The authors do include some discussion on tumor grade (which is only partially correlated to tumor fraction), but it would be important to address the effect of tumor fraction on the classification performance, for instance by in silico experiments where tumor signal is diluted by healthy control data.

- Authors should elaborate on the intended use case, this is essential to gauge the value of the method. For example, classifiers are applied at 99% specificity; is that good enough for screening, where the vast majority of the cases will be negative? In other words: 1% false positives is a problem if you expect 0.1% true positives in a population of 100.000 people, but good enough if you preselect patients who are suspected to have cancer in the first place and where you (for instance) expect 20% positive. On this topic, the authors should perhaps include not only the AUCs but also the sensitivity at a certain specificity.

- Over multiple experiments, there is a consistently higher performance in testing vs training. This is very counterintuitive and even a bit suspicious. Related: how are the 95% CIs calculated? They are very high, esp since there are only 19 ESCA samples in the test set (Fig 5A). While I think it is a good idea to pre-split the data in train and test cohort once (and before any analysis on the data) and then report the performance on the test, to estimate variability it would be good to include either the cross validation, or the results from multiple splits in train/test. Is the test performance indeed systematically higher than the train performance across these multiple splits or across CV?

- The multi-cancer early detection ('CSO classifier') is too unclear. Some more introduction of the 18 clusters is appropriate. It also remains unclear why the ATAC-seq overlap is even necessary. Why not just perform multi-class classification on the cases in the cfDNA data? At the very least this should be

motivated, and arguably even compared. A multi-class classifier (RF) was used. Did the authors compare to a 1-vs-rest approach?

- The models make use of SVMs, LRs and RF machine learning models. What is the rationale for using these different models and why are different models used on different occasions? If there is any experimental results behind this, this should be reported. If there is a rationale, this should be described.

- There is mention of a 20-fold cross validation (r488) for the THEMIS biomarker integration model. It is unclear on what fold of the data this is performed. Given that the individual biomarker models are already trained, it seems appropriate to use nested cross-validation to prevent re-use of the same training data in training of the individual models and integrative model.

- The data is randomly subsampled to 60M reads (presumably per sample; r98). Why is this done? What is the effect of this on the results? How robust is the performance to variation in sequencing throughput?

- r125: would be good to specify that matched refers to WGS of the same cfDNA sample (and not of the tumor).

- tSNE is a dimension reduction algorithm and not a clustering algorithm. It does not give clusters, but is merely a way of showing high dimensional data in 2 (or more) dimensions. It is true that clusters may then become apparent visually, but no clustering is done in the algorithm whatsoever.

- hg19 used, performance may be better with Chm13.

- The authors postulate that CNV-positive regions shed more cfDNA influencing the MFR signal. Could this be corrected, causing the MFR and CNA profiles to become more independent and complementary?

- r212 LR is undefined at this point in the text (is later defined in the Methods).

- The results are hard to read as is. It would be better if some intuitive and compact version of the methods is included in the results, and the methods section is used for details that are required for reproducing the results. This will make the manuscript much easier to follow and the figures much easier to understand. For instance the whole machine learning and THEMIS score is reduced to 2 sentences (r113-r117). The section r142-r144 is too succinct to follow. The text describing the results in Fig 3 does not really help to understand without going back and forth to the methods.

- The grammar is a bit awkward at times and can use some polishing. Non exhaustive examples: r26: "analysis of cell-free methylation, fragmentation, and copy number alteration (CNA) characteristics" Would be more comprehensible if phrased as: "analysis of methylation, fragmentation, and copy number alteration (CNA) characteristics from cell-free DNA." r44. "deadly disease and responsible". r53. "genetic alterations such as somatic mutation is a hallmark of cancer". r61. "to improve detection limit over".

- Figure 3 small and difficult to interpret; red is not in the legend, no x or y labels. Why are only COREAD samples used? How is the y-axis sorted in Fig 3A-C?

- Figure 4: TSNE is not clustering very well, yet phenomenal classification performances are reported, can this be interpreted? For instance, can only the test samples be visualized and false positives and false negatives be indicated? Difficult to conclude that THEMIS is better than individual markers.

Reviewer #3 (Remarks to the Author): expertise in cfDNA genomics and bioinformatics

NCOMMS-22-34997

Summary:

Bie, Tan, Gao and colleagues present an approach for early-stage cancer detection using genome-wide methylation sequencing of cell-free DNA (cfDNA). This platform integrates the analysis of DNA methylation, fragmentation, and copy number alterations (CNA) features extracted from NEBNext enzymatic methyl-seq (EM-seq). The authors processed these cfDNA features across genome bins and first tested each independently, then followed by an integrated classifier using regularized logistic regression. The applications demonstrated using this proof-of-concept focused on cancer detection and tissue-of-origin classification. Initial testing was applied using smaller cohorts of 20 colorectal cancer samples for methylation analysis and 270 pan-cancer patients for CNA and fragmentation profiling. For testing on a larger cohort, the authors used data from 780 cancer patients and 497 healthy controls, whereby a training and test set was split from the start at 7:3 ratio. The cancer detection performance was highest for THEMIS, which had an AUC of 0.971 for cancer detection. For tissue-of-origin classification, the performance of THEMIS varied between 50-66% accuracy. The sensitivity, in particular for stage I disease is high.

Overall, the manuscript is well-written, and the results are clearly presented. cfDNA methylation is now a popular mainstay in liquid biopsies for cancer applications. Assays that allow for the most tumor-specific information to be extracted from the limited amounts of cfDNA in the blood is paramount. However, there are some major concerns with the technical novelty, machine learning performance framework, and data availability that limit the enthusiasm for this study.

1. It has been shown in past studies that this is a promising avenue, including cfTAPS (Science Advances, 2021; PMID:34516908) and cfNOMe (Genome Medicine, 2020; PMID:32580754; authors did not cite). Both methods perform multimodal profile methylation, fragmentation patterns, and copy number alterations. In particular, cfNOMe also used NEBNext EM-seq. It is unclear how the methodologies described in the current study is a new advance over existing methods because there no comparison was performed. The algorithms described here for cfDNA feature extraction are derivative or use existing approaches—sometimes from methods published over a decade ago. The approach of THEMIS

to integrate the feature modalities is quite rudimentary with very limited characterization or inspection of its results, such as presence of co-linearity, etc, for the regression. Collectively, this limits the novelty of the approach.

2. The datasets described in this manuscript is a major strength of the manuscript. However, the Data Availability Statement states that these data are not available for download without a request from the corresponding author.

3. Concordance of CNA and fragmentation patterns is based only on a single patient sample, which does not address the variability in the data between patient samples. The chromosomes used in the plasma aneuploidy (PA) score was not clearly defined and if it is indeed the top 5 chromosomes within each sample, then there may be variability between samples. Also, what is the interpretation of 0 and 1 for FSI (GC-corrected)?

4. In Figure 3D, cfDNA CNA and FSI features initially selected/normalized using the healthy controls (FSI using Wilcoxon test and CNA used controls for Z-score). Therefore, the observation that there were no signals in the healthy controls seems obvious.

5. The authors should include analysis and comparison of the ctDNA fraction. While early-stage cancers tend to have lower ctDNA fraction, it is not always the case. A careful comparison of the fraction will add an important layer to the assessment of the platform.

6. Training and testing framework with the MONITOR study cohort requires some clarification and adjustments.

a. The hold-out is not a true independent external validation. Was the 7:3 split stratified such that all numbers are roughly equal? To avoid the 'jackpot' split and to capture the potential variability in the cohort groupings, the authors should consider bootstrapping (with sample replacement) for this partitioning strategy.

b. Are there any biases in the sample collection and data quality between the 6 hospitals? The authors can show this using PCA/tSNE and indicate the hospital source for healthy controls and each cancer type, independently.

c. Were the Principal Components (PCs) chosen for each fold or from the whole training set? Ideally, for the training performance, the selection of PCs as features should be included into the cross-validation strategy (PCs selected for each fold, then validated on the hold-out).

d. This should also be the case for the 20-fold CV for THEMIS—for each fold: (1) the PCs should be selected from scratch with 10-fold CV, (2) probabilities generated (MFR, FSI, FEM, CAFF), (3) then fed into the ensemble model. (For the final model applied to the test data, the whole training set can be used.)

e. How were confidence intervals for AUCs computed?

7. The term biomarker refers to some characteristic *evaluated* as an indicator of a medical state and is accurately and reproducibly measured. In the context of this manuscript, the term 'biomarker' used throughout is misleading. The authors should consider another term, such as 'feature', 'data modality', 'metric', 'measurement'.

8. The study falls short in Figure 6, since the tissue-of-origin determination could likely be substantially improved. Instead of ATAC data, they should also use available methylation data from tissues, which would likely improve the accuracy of tissue-of-origin determination. Also, there should be a discussion on the modest performance of this analysis.

Reviewer #4 (Remarks to the Author): expertise in DNA methylation analysis

In this manuscript, Gao and colleagues proposed THEMIS, a machine learning approach for early cancer detection and classification from cell free DNA. The concept is not novel with Galleri, the commercial product available from GRAIL. The TET-APOBEC approach is also commercially available. Therefore, the major improvement has to be the improved performance (prediction accuracy). When taking at the face value, the reported accuracy does outperform that from GRAIL. However, there are many arbitrary design options in the model without careful justification and may lead into overfitting concerns.

Major comments:

1) For a AI model to be clinically useful, it needs to be robust (say from different processing lab etc). Although this manuscript reported a cohort from 6 different hospital, they were pooled together before randomly assigning 30% as testing. Therefore, there is no strong batch difference between the training dataset and the testing dataset. However, in clinical application, the sample was newly collected and may have latent batch difference that may affect the sample processing and data acquisition, which may in turn negatively influence the performance. Therefore, I will suggest the authors to validate their model in a new, independent cohort. Alternatively, they could select patients from 4 hospital as training and those from the remaining 2 hospitals as testing.

2) There are many arbitrary design options throughout the model, which further boosts the overfitting suspicion. Here is just a few examples:

- a. The authors reported hyperparameter tuning using 13 bootstraps. Why 13 instead of 10, 20 or 30?
- b. The normalization scheme is largely inconsistent. For example, no normalization for MFR, z-scoring for FSI and quantile normalization against healthy controls for FEM. Why?
- c. Different machine learning models were used for different models. Moreover, why not a single model that combines all features?

Reviewer #1 (Remarks to the Author): expert in DNA fragmentation analysis

The author utilized multi-modality information from the enzymatic conversion based low-coverage whole-methylome sequencing for the multi-cancer early detection. Overall, it is an interesting study. However, I have some major concerns of the study:

1. Downsample experiment, what is the minimum required sequencing depth for each modality and what is that for the integrate multi-modality? Maybe a downsampling experiment is needed.

Author response:

We are thankful for the reviewer's suggestion for a downsampling experiment to investigate the minimum sequencing depth required for feature extraction and model development. In fact, we investigated this question in our pilot analysis with 773 MONITOR samples (including 306 healthy controls and 467 cancer samples) sequenced at high depth, and randomly downsampled the aligned bam files to 120M, 90M, 60M, 30M, 15M, and 3M paired reads. At each depth, we kept the original sample cohort assignment (training or test) and re-trained the same integrative THEMIS model. First, we observed an inverse relationship between the sequencing depth and the number of principle components required to explain the variance of MFR, FSI, and FEM features, which indicates that lowering sequencing depth decreases the signal-to-noise ratio for feature extraction. Given this observation, it is unsurprising that models for MFR, FSI, FEM, and THEMIS trained with data of lower sequencing depth manifested overfitting issues and CAFF showed reduced prediction power at lower depth (e.g., 3M and 15M reads). With input data of 60M reads, both tier-1 and tier-2 (integrative) models achieved performance comparable with data of higher depth and showed no apparent overfitting. Therefore, we chose 60M reads as the optimal sequencing depth and developed everything based upon this depth. Again, we thank the reviewer's suggestion and have added this result to the Results section (Lines 328 - 339 and Figure S10).

2. Need to share the raw data and fragment-level data. There is no way that other people can replicate their study. "available upon request" usually means data is not available to the community.

Author response:

We understand that the data used in the manuscript is valuable for the scientific community to replicate this study and apologize for not providing an access link. WMS genomic sequencing files of the MONITOR cohort have been deposited in GSA for Human with accession no. HRA003209 (<https://ngdc.cncb.ac.cn/gsa-human/s/Cr8mdP84>). The extracted feature data for model development, including bin-level MFR and FSI data, chromosome-level CAFF data, and the FEM data have been deposited in the OMIX, China National Center for Bioinformation with accession no. OMIX002936.

We have also deposited the code to reproduce the figures in this manuscript in GitHub (<https://github.com/hchyang/themis.git>), which is publicly available.

3. In the multi-modality classification, which modality contributes most? Need to show the fraction of power from each modality in the ensemble model.

Author response:

A major strength of our cfDNA WMS data is the ability to simultaneously profile multiple genomic features and we thank the reviewer for pointing out this important question regarding the contribution of each modality. THEMIS used a logistic regression (LR) model to integrate modalities and the beta coefficients for MFR, FSI, CAFF, and FEM are 1.35, 1.15, 3.25, and 1.59 respectively, which may represent the relative importance of each modality for the ensemble model. Specifically, CAFF contributed the most, which may result from the high specificity of chromosome instability in cancer detection. We listed modality coefficients in the Methods section and have added discussions on feature importance in the Results section (Lines 311 - 314).

4. When assessing the concordance between WGS and WMS, do they have similar sequencing depth? Only one pair of sample is shown, not sure how stable it is, the author needs the replicates.

Author response:

We apologize for insufficient clarification on the concordance assessment between WGS and WMS. Indeed, both types of data were randomly downsampled to 60 million reads to ensure similar genome coverage and we have added this specification in the Results section (Lines 153 - 154). To visualize fragmentation and copy number profiles across the genome, we showed one sample as an example in Figures 2A and 2C and have added 14 more samples (two per cancer type) in Figure S2 to visually demonstrate the similarity between WMS and WGS. In addition, we quantified the cross-platform similarity of copy number and fragmentation for the entire cohort of 490 samples in Figures 2B and 2D. In Figure 2B, chromosome arm-level aneuploidy of each sample was quantified by PA score (Leary et al. 2012) and the PA scores of WGS against WMS data were plotted. In Figure 2D, genome-wide fragmentation profile of each sample was proxied by its Pearson correlation coefficient (PCC) with the healthy control reference profile because healthy individuals had highly similar fragmentation patterns (Cristiano et al. 2019). The PCC calculated with WGS data against the PCC calculated with WMS data for each sample was then plotted.

5. The title is about multi-cancer early detection. However, only ~30% of samples are early-stage. This is misleading.

Author response:

We agree with the reviewer that this study recruited less early-stage than advanced-stage samples. However, we analyzed the predictive power of THEMIS for individual

stages to assess its performance for early cancers. For example, Figure S11B illustrated the distribution of THEMIS scores for each stage and there was a conspicuous difference between early-stage (I and II) samples and healthy controls. In Figure 5B, we showed the sensitivity of detection for each cancer type at each stage with 99% specificity and there was adequate sensitivity for early-stage samples. We are aware that the detection performance for early-stage patients should be interpreted with caution given the limited sample size and have added this discussion in the Discussion section (Lines 554-556). To assess the clinical performance of THEMIS more precisely, prospective studies with larger sample size and balanced stage distribution are ongoing.

For Figure 4A, the author need to separate the stage and show it again. I suspect the separation is mainly just in late-stage. Otherwise, the description might be misleading to the reader that early-stage signals here are also clearly distinguished between cancer types.

Author response:

We appreciate the reviewer's suggestion for testing whether cancer types could be separated by weak early-stage signals. As the reviewer suggested, we have separated cancer samples of Figure 4A into early (I and II, n = 272) and late (III and IV, n = 375) stages and plotted tSNE visualizations for the two groups separately in Figure S4. We agree with the reviewer's hypothesis that cancer type separation is more pronounced in late-stage samples, which is not surprising given the extremely low concentration of circulating tumor DNA fragments in early-stage patient blood. However, the weak cancer type specificity in early-stage cancer samples should not be confused with the high performance in cancer detection, since we can observe the separation of early-stage cancer samples from healthy controls even in these two-dimensional tSNE plots. The dimension of feature space utilized by THEMIS model is much higher and may enable more sensitive detection of weak cancer signals. We have added this result in the Results section (Lines 272 - 276).

6. The best tuned elastic-net mixing parameter alpha. Did they tune the parameter based on the test cohort or cross validation within the training cohort? This is not clearly described.

Author response:

We apologize for missing the description of hyperparameter tuning for the integrative model. We have added in the Methods section that we tuned the hyperparameters for the final elastic-net LR model by cross validation within the training cohort (Lines 726 - 727).

Minor:

1. How is the bisulfite conversion rate of the WMS data?

Author response:

We apologize for leaving out this important information for DNA methylation

sequencing. Indeed, we spiked-in unmethylated lambda DNA to estimate the conversion rate of unmodified cytosine. The 1,277 cfDNA samples had a median conversion rate of 99.4%. We have added the conversion rates in the Results section (Lines 109 - 111) and Figure S1. It is also to be noted that to reduce any potential influence on MFR profiling by low conversion rate, we estimated the conversion rate of each fragment by calculating the fraction of converted non-CpG cytosine and discarded fragments with a conversion rate below 95% as described in the Methods section.

2. 4 ml of plasma is a lot. Don't have to emphasis on it in the abstract...

Author response:

THEMIS used 4 ml of plasma isolated from 10 ml of whole blood, which is the standard volume of a blood collection tube for liquid biopsy applications. In fact, this amount is lower than previously published large-scale multicancer detection and localization studies: 20 ml of whole blood in the CCGA study conducted by GRAIL (Liu et al. 2020) and ~7.5 ml of plasma by CancerSEEK (Cohen et al. 2018). However, we totally agree with the reviewer that 4 ml of plasma is still a lot and have removed the emphasis on it from the abstract. Technically, it is possible to perform whole-methylome sequencing with even lower cfDNA input than that used in the current study, which is currently under active investigation by our group.

Reviewer #2 (Remarks to the Author): expertise in machine learning methods

Bie et al present a methylation + machine learning method called THEMIS, which enables cancer detection from multimodal cfDNA signals. Their study includes application of enzymatic methyl-seq (EM-seq), which was recently published as an alternative to bisulfite sequencing, to a total of 1277 plasma cfDNA samples at ~2x coverage. These data are subjected to various machine learning analyses, integrating methylation, fragmentation, copy number and fragment-end motif information, to show very good performance separating cancer from healthy control samples and feasibility to detect the origin of the cancer. The work has a lot of merit and I consider it a worthy addition to the rapidly developing field. Some concerns need to be address / clarified in order to make it suitable for publication:

1. Data and code availability - the authors should better adhere to the data availability requirements of the journal (and of modern research to be honest). The manuscript would be much more valuable if data are deposited in a (controlled access) repository with clear guidelines on the criteria for access (and not only available upon reasonable request). Same holds for the code and scripts (machine learning, sequencing data processing, etc), for which there is no mention on how it is available for the purpose of reproducing the results in the manuscript and for utilisation by the scientific community. I read that there is a patent filed based on this work. That is absolutely fine, but this should not (and does not have to) conflict with non-commercial (academic) reproducibility and reuse of this work.

Author response:

We understand that data and code availability is essential for the reproducibility and reuse of this work by the scientific community. We have deposited the cfDNA WMS raw data, the extracted features, and the code to reproduce all figures in this manuscript in publicly available repositories and please see our response to Comment #2 from Reviewer 1 for the links. Because the algorithms embedded in the scripts for sequencing data processing and analysis are under active patent filing, at this point we can only provide relevant scripts upon direct correspondence from the academic community.

2. I think there should be more discussion around tumor fraction. The authors do include some discussion on tumor grade (which is only partially correlated to tumor fraction), but it would be important to address the effect of tumor fraction on the classification performance, for instance by in silico experiments where tumor signal is diluted by healthy control data.

Author response:

We would like to thank the reviewer for the insightful suggestion for using tumor fraction to evaluate classifier performance. Instead of in silico experiments, we intend to leverage the concept of clinical limit of detection (cLOD), a measure recently created by Jamshidi et al (Jamshidi et al. 2022), to investigate the clinical

performance of THEMIS. We have calculated the mean variant allele frequency (VAF) of 65 plasma samples in our cohort whose paired tumor tissue and white blood cell (WBC) were available and single nucleotide variant (SNV) was detected in both tissue and cfDNA after WBC background removal. With some modifications from the original Jamshidi et al study, in our analysis cLOD has been defined as the mean VAF at which the probability of positive cancer signal detection is at least 50% with a 99% specificity. We have noted that the cLOD of THEMIS (7.3×10^{-4} mean VAF) is lower than individual modalities, in agreement with our hypothesis that feature integration can boost detection sensitivity. The cLOD of individual modalities and the integrative THEMIS classifier has been shown in Figure S12 and we have added this result in the Results section (Lines 355 - 364). Detailed methods for cLOD calculation have been added to the Methods section (Lines 738 - 746).

3. Authors should elaborate on the intended use case, this is essential to gauge the value of the method. For example, classifiers are applied at 99% specificity; is that good enough for screening, where the vast majority of the cases will be negative? In other words: 1% false positives is a problem if you expect 0.1% true positives in a population of 100,000 people, but good enough if you preselect patients who are suspected to have cancer in the first place and where you (for instance) expect 20% positive. On this topic, the authors should perhaps include not only the AUCs but also the sensitivity at a certain specificity.

Author response:

We thank the reviewer for pointing out this critical question regarding the intended use of our method. As a blood-based multi-cancer test, THEMIS targets a general screening population with low to average risk of having cancer. For instance, the United States cancer incidence rate in 2019 is 0.44% for all ages combined and ~1.6% for people over 50 years old (SEER data, 2019). A multi-cancer detection test for this type of population requires a sufficiently high specificity to achieve a low false positive rate and high accuracy in CSO localization, so that informative diagnostic workups could be advised following a positive prediction. In this regard, the sensitivities reported in this manuscript (e.g., Figure 5B) were at a fixed specificity of 99% and the overall sensitivity was 86% in the test cohort. When applied to a population with a 1.5% incidence rate per year of cancer, our test would detect 1,290 true positives and result in 985 false positives per 100,000 participants, yielding a positive predictive value (PPV) of 57%. It should be noted that this manuscript is still a proof-of-concept methodology study with a relatively small sample size, thus the fixed specificity of 99% was arbitrarily pre-selected with the limited number of control samples in the training cohort. A prospective study with a much larger sample size is currently conducted by our group to investigate the sensitivity, specificity, and PPV of the THEMIS approach more precisely.

4. Over multiple experiments, there is a consistently higher performance in testing vs training. This is very counterintuitive and even a bit suspicious. Related: how are the 95% CIs calculated? They are very high, esp since there

are only 19 ESCA samples in the test set (Fig 5A). While I think it is a good idea to pre-split the data in train and test cohort once (and before any analysis on the data) and then report the performance on the test, to estimate variability it would be good to include either the cross validation, or the results from multiple splits in train/test. Is the test performance indeed systematically higher than the train performance across these multiple splits or across CV?

Author response:

We appreciate the reviewer's insightful suggestion for multiple train/test splits to estimate model variability and have performed 100 random splits of the dataset into training and test cohorts at the fixed ratio of 7:3 before any analysis. Feature extraction and model training are then implemented for each split, and AUCs for individual modalities and the integrative THEMIS model have been summarized in Figure S14. We find that both training and testing performances are highly consistent among the 100 runs for all modalities as well as the final model. We also note that testing AUCs are slightly lower than (but still on par with) training AUCs for all modalities except CAFF. These results demonstrate the stability of our method and suggest that the slightly higher performance in testing vs training, as the reviewer pointed out, probably resulted from sample split. We have added discussions on model variability in the Discussion section (Lines 507 - 512).

We also apologize for missing the calculation method of 95% CI. The CI of AUC (Figures 4B and 5A) was calculated with 2000 stratified bootstrap replicates (Robin et al. 2011). This has been added to the relevant figure legends.

5. The multi-cancer early detection ('CSO classifier') is too unclear. Some more introduction of the 18 clusters is appropriate. It also remains unclear why the ATAC-seq overlap is even necessary. Why not just perform multi-class classification on the cases in the cfDNA data? At the very least this should be motivated, and arguably even compared. A multi-class classifier (RF) was used. Did the authors compare to a 1-vs-rest approach?

Author response:

We sincerely apologize for the insufficient introduction of ATAC-seq clusters and their role in cancer type classification. We would like to break down this comment into three points and respond to each one.

I. Why not just perform multi-class classification on the cases in the cfDNA data?

We assume that the reviewer suggests running multi-class classification based on the same genomic features as those for cancer detection, i.e., MFR, FSI, CAFF, and FEM. We in fact did try this approach and obtained worse multi-class classification results (data not shown). However, the result is not surprising given that these cancer detection features are genome-wide and have been shown to manifest little cancer-type specificity. For example, the methylation level, from which MFR is derived, has been found to be predominantly hypomethylated in most cancer types

(Chan et al. 2013; Ushijima 2005). Similarly, FSI profiles the fragmentation size distribution of 5-Mb genomic windows, which also showed limited cancer type specificity in the original Cristiano et al paper (Cristiano et al. 2019). It is likely that the large window size in calculating MFR and FSI (large size due to low-depth data) obscures cancer-type specific variation in smaller genomic regions. FEM analyzes global changes in cfDNA end motif patterns, which was proposed to associate with alterations in the levels of relevant nucleases in cancer patients (Han et al. 2020). Finally, CAFF quantifies copy number changes of chromosome arms but multiple cancer types may share "hotspot" chromosomal changes. Based on these biological mechanisms, the multi-class classification result based on genome-wide features is worse than that based on cancer type specific regions.

II. Why the ATAC-seq overlap is necessary?

As explained in response to Point #1, we sought to focus on cancer type specific genome regions to extract epigenetic features for multi-class classification. It is well established that cis-regulatory DNA elements such as promoters and enhancers regulate gene expression, and recent studies have further identified the roles of enhancers in controlling the transcriptional networks to determine tumor subtypes (Chen and Liang 2020; Zhang et al. 2019). Because chromatin accessibility is a proxy for the activity of regulatory elements, Corces et al profiled the chromatin accessibility landscape for 23 types of primary tumor samples in TCGA by ATAC-seq and annotated 18 ATAC-seq clusters by cancer tissue specificity (Corces et al. 2018). In contrast to direct modeling with plasma cfDNA data originating from a mixture of contributing tissues, this orthogonal data type identified genuine cancer specific putative distal enhancers with which we extracted methylation and fragmentation profiles for cancer type classification. As expected, we observed lower methylation as well as lower fragment coverage in relevant ATAC-seq clusters with our cfDNA data (Figure 6A), supporting the tumor relevance for the features we employed for CSO classification. We have elaborated the introduction on chromatin accessibility and 18 ATAC-seq clusters in the Results Section (Lines 391 - 397).

III. Multi-class classifier vs 1-vs-rest approach.

We thank the reviewer for raising this point. It is tempting to build a K-class classifier by combining K 1-vs-the-rest binary classifiers, K being equal to the number of the class labels. However, this leads to some well-known difficulties (Duda and Hart 1973). For example, this figure (<https://i.stack.imgur.com/qUP9Y.png>), showing the classification result of four 1-vs-the-rest classifiers, demonstrates a region of input space that is ambiguously classified. We chose a multi-class classifier to avoid these difficulties.

6. The models make use of SVMs, LRs and RF machine learning models. What is the rationale for using these different models and why are different models used on different occasions? If there is any experimental results behind this, this should be reported. If there is a rationale, this should be described.

Author response:

We thank the review for suggesting us to explain the rationale for model selection in the manuscript and have now added explanations in the Methods section for each model (Lines 719 - 722, 727 - 729, and 768 - 769). For modalities (MFR, FSI, FEM) with relatively large dimensionality relative to the sample size, machine learning methods were applied after dimensionality-reduction by PCA. For these modalities, all three mentioned methods (SVM, LR, RF) were applied, and we chose the one with the best performance (i.e., SVM for MFR and FSI and LR for FEM). In tier 2 of the entire model, the model input is a vector of four output values from four tier-1 models. Given its relatively small size, we chose to apply the parametric statistical model (LR) to better interpret the contribution from each modality.

For the multiclass CSO classification, we chose RF which is known to be less prone to overfitting, because the number of features (54) is relatively large. Indeed, we tested the multinomial LR for the CSO classification and observed severe overfitting with the training accuracy of 80% and the testing accuracy of 46%.

7. There is mention of a 20-fold cross validation (r488) for the THEMIS biomarker integration model. It is unclear on what fold of the data this is performed. Given that the individual biomarker models are already trained, it seems appropriate to use nested cross-validation to prevent re-use of the same training data in training of the individual models and integrative model.

Author response:

We sincerely apologize for the confusion about the training method for the integrative THEMIS model. We pre-split the entire cohort into a training group and an independent test group before any analysis. The 20-fold cross validation for THEMIS model development was performed within the training group using the final prediction scores of individual modalities for each sample. That being said, the folds for THEMIS model training are independent of the folds for the training of individual modality models and therefore, we did not re-use the same training data in training the two tiers of models. We have modified the Methods Section to clarify the training strategy of the integration model (Lines 726 - 727).

8. The data is randomly subsampled to 60M reads (presumably per sample; r98). Why is this done? What is the effect of this on the results? How robust is the performance to variation in sequencing throughput?

Author response:

This comment is similar to Comment #1 from Reviewer 1 and please check our response there for the evaluation of feature stability and model performance at various sequencing depths. The results shown in the manuscript were analyzed with all samples subsampled to 60M reads, a relatively low sequencing depth to control the cost of the assay while maintaining similar performance as higher depths.

9. r125: would be good to specify that matched refers to WGS of the same

cfDNA sample (and not of the tumor).

Author response:

We agree that this specification should be made and have revised this line and the line above to "...evaluate copy number and fragmentation profiling from cfDNA WMS data and assessed their concordance with WGS data of similar sequencing depth (60M reads) generated with the same cfDNA samples. "

10. tSNE is a dimension reduction algorithm and not a clustering algorithm. It does not give clusters, but is merely a way of showing high dimensional data in 2 (or more) dimensions. It is true that clusters may then become apparent visually, but no clustering is done in the algorithm whatsoever.

Author response:

We sincerely apologize for this incorrect description of the method and in the manuscript tSNE was used for data visualization instead of clustering. We have revised the original description of tSNE results to "Visualization of all four individual genomic features by tSNE plots showed clear separation of cancer patients from healthy controls, supporting their utility for cancer detection. Moreover, cancer samples of the same types appeared to be separable by methylation, fragmentation size, and CNA profiles, suggesting that these features may have cancer type-specific signatures despite the arbitrary segmentation of the genome for MFR and FSI profiling" (Lines 260 - 272) and corrected the title for tSNE method in the Methods Section (Line 684).

11. hg19 used, performance may be better with Chm13.

Author response:

We are very grateful for this suggestion and agree that the advent of more complete genome assemblies such as Chm13 may help improve model performance. However, given the huge size of datasets analyzed in this study and the timeframe of this manuscript, re-analyzing the entire dataset with a new reference genome is beyond the scope of this manuscript, which mainly aims to demonstrate the proof of principle for multimodal analysis of WMS data for cancer detection. We will investigate the performance with Chm13 as well as the graph reference genome in the future to further refine our approach.

12. The authors postulate that CNV-positive regions shed more cfDNA influencing the MFR signal. Could this be corrected, causing the MFR and CNA profiles to become more independent and complementary?

Author response:

We apologize for the confusion about this statement and would like to better clarify the associations between MFR and CNA here. MFR was calculated as the ratio of methylated fragments within a window and therefore was not affected by the total number of fragments. In principle, the methylation level of a genome is associated

with many biological factors including but not limited to CNA. Although associations between local methylation changes and copy number alterations in tumor cells have been identified, these two events also have distinct mechanisms and can thus provide complementary cancer signals. Our postulation that CNV-positive regions shed more cfDNA influencing the MFR signal referred to the fact that in blood these regions had higher fractions of tumor-derived cfDNA, which was generally hypomethylated compared with normal cfDNA, and as a result cancer signals (i.e., hypomethylation) were stronger in these regions than CNA-negative regions.

13. r212 LR is undefined at this point in the text (is later defined in the Methods).

Author response:

We are sorry for this negligence and have changed LR to logistic regression in this line.

14. The results are hard to read as is. It would be better if some intuitive and compact version of the methods is included in the results, and the methods section is used for details that are required for reproducing the results. This will make the manuscript much easier to follow and the figures much easier to understand. For instance the whole machine learning and THEMIS score is reduced to 2 sentences (r113-r117). The section r142-r144 is too succinct to follow. The text describing the results in Fig 3 does not really help to understand without going back and forth to the methods.

Author response:

We appreciate the reviewer's helpful advice on including informative methods in the results section for easier understanding. As pointed out by the reviewer, we have added descriptions of machine learning methods for individual modalities and the integration model in the overview of THEMIS approach (Lines 131 - 138). We have also elaborated on the method of FSI quantification to specify that we measured the similarity of each sample to the reference profile generated from healthy controls for each platform (Lines 180 - 183). In addition, we have added more details about the quantification methods and data interpretations for Figure 3 (Lines 192 - 215). We hope these revisions can strengthen the readability of this manuscript.

15. The grammar is a bit awkward at times and can use some polishing. Non exhaustive examples: r26: "analysis of cell-free methylation, fragmentation, and copy number alteration (CNA) characteristics" Would be more comprehensible if phrased as: "analysis of methylation, fragmentation, and copy number alteration (CNA) characteristics from cell-free DNA." r44. "deadly disease and responsible". r53. "genetic alterations such as somatic mutation is a hallmark of cancer". r61. "to improve detection limit over".

Author response:

We are sorry about the grammar problems and have done language polishing in the

revision. Specifically to the examples pointed out by the reviewer, we have rephrased “analysis of cell-free methylation, fragmentation, and copy number alteration (CNA) characteristics” to “analysis of methylation, fragmentation, and copy number alteration (CNA) characteristics from cell-free DNA”. We have changed “deadly disease and responsible” to “deadly disease, accounting for”, “genetic alterations such as somatic mutation is a hallmark of cancer” to “genetic mutation is a hallmark of cancer”, and “Characterizing the vast number of plasma epigenetic changes is expected to improve detection limit over searching point mutations” to “Compared with searching point mutations, characterizing the vast number of plasma epigenetic changes is expected to improve detection sensitivity”. We hope the revised manuscript has improved grammar and readability. If considered necessary by the editor or reviewers, we will consult a professional language editing service for more advice.

16. Figure 3 small and difficult to interpret; red is not in the legend, no x or y labels. Why are only COREAD samples used? How is the y-axis sorted in Fig 3A-C?

Author response:

We thank the reviewer for this valuable feedback and have added relevant labels and legend to Figure 3 for better interpretability. Specifically, we have added red in the legend, and added x label (“Plasma samples”) and y label (“1,795 1-Mb windows”) to the heatmaps in Figures 3A-C. The y-axis was sorted by the chromosomal coordinates of the windows from chromosome 1 to chromosome 22, and we have added this description to the figure legend. Because this analysis only aimed to demonstrate the proof of principle that cfDNA WMS data could provide complementary multimodal cancer signals, we used a small cohort of COREAD samples as pilot examples for illustration. If desired, other cancer types can be used as well and we expect to make similar observations.

17. Figure 4: TSNE is not clustering very well, yet phenomenal classification performances are reported, can this be interpreted? For instance, can only the test samples be visualized and false positives and false negatives be indicated? Difficult to conclude that THEMIS is better than individual markers.

Author response:

We are grateful for this insightful question about the discrepancy between visualization effect and model performance. It should be noted that the data of each individual marker had high dimensionality (e.g., 1,846 for MFR and 502 for FSI) and to reduce data dimensionality for visualization, tSNE plots in Figure 4A displayed samples on a two-dimensional space. Therefore, compared with machine learning models which utilized more data dimensions, loss of information is expected in tSNE plots, especially for early-stage cancer samples with weak signals. In addition, these markers provide complementary information (as shown in Figure 3 and Figure 4C) and the strength of THEMIS is the integration of predictions from individual markers for better performance.

Reviewer #3 (Remarks to the Author): expertise in cfDNA genomics and bioinformatics

Summary: Bie, Tan, Gao and colleagues present an approach for early-stage cancer detection using genome-wide methylation sequencing of cell-free DNA (cfDNA). This platform integrates the analysis of DNA methylation, fragmentation, and copy number alterations (CNA) features extracted from NEBNext enzymatic methyl-seq (EM-seq). The authors processed these cfDNA features across genome bins and first tested each independently, then followed by an integrated classifier using regularized logistic regression. The applications demonstrated using this proof-of-concept focused on cancer detection and tissue-of-origin classification. Initial testing was applied using smaller cohorts of 20 colorectal cancer samples for methylation analysis and 270 pan-cancer patients for CNA and fragmentation profiling. For testing on a larger cohort, the authors used data from 780 cancer patients and 497 healthy controls, whereby a training and test set was split from the start at 7:3 ratio. The cancer detection performance was highest for THEMIS, which had an AUC of 0.971 for cancer detection. For tissue-of-origin classification, the performance of THEMIS varied between 50-66% accuracy. The sensitivity, in particular for stage I disease is high.

Overall, the manuscript is well-written, and the results are clearly presented. cfDNA methylation is now a popular mainstay in liquid biopsies for cancer applications. Assays that allow for the most tumor-specific information to be extracted from the limited amounts of cfDNA in the blood is paramount. However, there are some major concerns with the technical novelty, machine learning performance framework, and data availability that limit the enthusiasm for this study.

1. It has been shown in past studies that this is a promising avenue, including cfTAPS (Science Advances, 2021; PMID:34516908) and cfNOMe (Genome Medicine, 2020; PMID:32580754; authors did not cite). Both methods perform multimodal profile methylation, fragmentation patterns, and copy number alterations. In particular, cfNOMe also used NEBNext EM-seq. It is unclear how the methodologies described in the current study is a new advance over existing methods because there no comparison was performed. The algorithms described here for cfDNA feature extraction are derivative or use existing approaches—sometimes from methods published over a decade ago. The approach of THEMIS to integrate the feature modalities is quite rudimentary with very limit characterization or inspection of its results, such as presence of co-linearity, etc, for the regression. Collectively, this limits the novelty of the approach.

Author response:

We sincerely appreciate the reviewer's insightful comment on recent technical advances in the field and apologize for insufficient justifications for the novelty of the THEMIS approach. As pointed out in the comment, multimodal profiling of cfDNA

epigenetic features is a promising avenue for non-invasive cancer detection, but currently there are only limited publications available including cfTAPS (Siejka-Zielińska et al. 2021). The reviewer also mentions cfNOME which was developed with EM-seq (Erger et al. 2020), but this approach was not intended for cancer applications. Compared with these two methods, the THEMIS approach presented in the current study demonstrates several major differences and advancements:

I. Although both TAPS and EM-seq are bisulfite-free methods for methylation sequencing, the TAPS protocol employs a pyridine borane (PyBr) reduction step to convert 5caC to DHU, which requires a lengthy 16-hour incubation. It is likely that experimental conditions of this step have to be carefully controlled to ensure efficient chemical reaction, while the enzymatic reactions of EM-seq are anticipated to be more robust. Besides technical differences, cancer detection strategies are also distinct between cfTAPS and THEMIS. As described in the cfTAPS paper, a mean depth of 11.6X was sequenced for the whole genome to identify differentially methylated enhancers and promoters for cancer classification. In contrast, THEMIS sought to detect methylation and fragmentation changes of large genome segments and therefore, much lower sequencing depth (~2X) suffices to provide excellent performance. As suggested by Reviewers 1 and 2, we have performed a downsampling experiment to determine the minimal required sequencing depth (please refer to our response to Comment #1 from Reviewer 1 for more details). The substantially lower sequencing cost for the THEMIS approach makes it more practical to be eventually utilized for large-scale cancer screening in a clinical setting.

II. Thank you for mentioning cfNOME, which was somehow omitted in our literature search. cfNOME explored the potential of EM-seq for simultaneous cfDNA methylation and nucleosome footprinting analysis for patients with kidney diseases. However, unlike these patients who have adequate renal cfDNA in their blood (>1%), the fraction of tumor-derived cfDNA in early-stage cancer patients is much lower (can be <0.1%) and therefore, the applicability of cfNOME for blood-based cancer diagnostics remains to be investigated. Our current study aims to develop and validate a sensitive multicancer early detection approach by optimizing and integrating multiple established circulating cancer markers, which is the first study to apply cfDNA EM-seq to a large clinical cohort and demonstrate the promise of multimodal profiling of methylation and fragmentation in improving the performance of cancer detection and localization. Similar to cfTAPS, cfNOME also requires relatively high sequencing depth for reliable nucleosome footprinting analysis and thus its potential for clinical use may be limited.

III. The sample size and patient diversity of both cfTAPS and cfNOME studies were rather limited. As a proof-of-concept study for cancer detection, cfTAPS included only 85 cfDNA samples (41 noncancer controls, 21 HCC, and 23 PDAC) for model development and 8 samples from an independent dataset (4 noncancer controls and 4 HCC) for validation. This small sample size may hinder proper evaluations of their model performance, especially in real-world clinical applications. In contrast, our study included 497 healthy controls and 780 patients of seven common cancer types and pre-split samples into a training and an independent testing cohort for model

development and testing. In addition, as suggested by Reviewer 4, we have validated the generalization capacity of THEMIS model in a new, independent validation cohort comprising 475 high-risk noncancer controls and 261 relatively early-stage cancer patients of five types. Although a prospective study in a larger cohort with complete patient long-term follow-up is still needed to investigate the real-world performance of THEMIS approach and establish its proper clinical utility, our data presented in the current manuscript has not only demonstrated the proof of principle for multimodal analysis of cfDNA WMS (EM-seq) data for multicancer detection and localization, but also carefully assessed our diagnostic model performance with a large number of clinical samples.

In summary, this current manuscript is the first investigation to apply EM-seq to blood cfDNA for multicancer early detection and localization, and the validity of this multimodal analysis approach was thoroughly evaluated on both technical and clinical perspectives. Therefore, the main scope of this manuscript is to validate the proof of principle for the clinical potential of THEMIS, and we have added comparisons with existing cfTAPS and cfNOMe methods in the Discussion section to better clarify the novelty of this manuscript (Lines 455 - 471). As pointed out by both Reviewer 1 and Reviewer 2, the multicancer cfDNA WMS datasets generated in this study are of great value for further mining, including the associations among different epigenetic features. These directions are currently under active investigation by our group, and we anticipate dissemination of the data in future manuscripts.

2. The datasets described in this manuscript is a major strength of the manuscript. However, the Data Availability Statement states that these data are not available for download without a request from the corresponding author.

Author response:

We have deposited the data in public repositories and please refer to our response to Comment #2 from Reviewer 1 for links to the data.

3. Concordance of CNA and fragmentation patterns is based only a single patient sample, which does not address the variability in the data between patient samples. The chromosomes used in the plasma aneuploidy (PA) score was not clearly defined and if it is indeed the top 5 chromosomes within each sample, then there may be variability between samples. Also, what is the interpretation of 0 and 1 for FSI (GC-corrected)?

Author response:

We apologize for these confusions about our methods used in evaluating concordance between WMS and WGS data. Please refer to our response to Comment #4 from Reviewer 1 for detailed descriptions of how we addressed sample variability in Figures 2B and 2D as well as the more examples we have added for visualization in Figure S2. The interpretation of FSI profiling by correlation coefficient has also been clarified in more details in the same response. When assessing chromosome CNA for each sample by PA score, the reviewer was correct that the top 5 chromosomes were selected on a per sample basis and were not necessarily

identical among samples. Nevertheless, they should remain highly similar between paired WMS and WGS data for late-stage patients with severe CNA, as illustrated in the example plots in Figures 2A and S2A.

4. In Figure 3D, cfDNA CNA and FSI features initially selected/normalized using the healthy controls (FSI using Wilcoxon test and CNA used controls for Z-score). Therefore, the observation that there were no signals in the healthy controls seems obvious.

Author response:

We are grateful to the reviewer for pointing this out. We have rephrased the interpretation of Figure 3D by deleting "In contrast, we identified no commonly altered windows for any biomarker in healthy controls".

5. The authors should include analysis and comparison of the ctDNA fraction. While early-stage cancers tend to have lower ctDNA fraction, it is not always the case. A careful comparison of the fraction will add an important layer to the assessment of the platform.

Author response:

We totally agree with the reviewer on the importance of ctDNA fraction analysis in assessing the performance of our approach. Please see Comment #2 from Reviewer 2 for detailed results and discussions.

6. Training and testing framework with the MONITOR study cohort requires some clarification and adjustments.

a. The hold-out is not a true independent external validation. Was the 7:3 split stratified such that all numbers are roughly equal? To avoid the 'jackpot' split and to capture the potential variability in the cohort groupings, the authors should consider bootstrapping (with sample replacement) for this partitioning strategy.

Author response:

We thank the reviewer for the insightful suggestion of bootstrapping to evaluate potential model variability. As also suggested by Reviewer 2, we have performed 100 random training/test splits at 7:3 ratio to investigate the variability of our machine learning models and please check our response to Comment #4 from Reviewer 2 for details. Each of the 100 splits as well as the split presented in the manuscript is stratified by cancer types and stages. In addition, we have included a new independent validation cohort to test our model performance and added the results and its discussions to the revised manuscript (Lines 365 - 384 and Figure S13), as suggested by Reviewer 4.

b. Are there any biases in the sample collection and data quality between the 6 hospitals? The authors can show this using PCA/tSNE and indicate the hospital source for healthy controls and each cancer type, independently.

Author response:

We thank the reviewer for this valuable suggestion. We have added tSNE plots of each feature for healthy controls and each cancer type in Figure S3 and no obvious batch effects among hospitals are observed.

- c. Were the Principal Components (PCs) chosen for each fold or from the whole training set? Ideally, for the training performance, the selection of PCs as features should be included into the cross-validation strategy (PCs selected for each fold, then validated on the hold-out).**
- d. This should also be the case for the 20-fold CV for THEMIS—for each fold: (1) the PCs should be selected from scratch with 10-fold CV, (2) probabilities generated (MFR, FSI, FEM, CAFF), (3) then fed into the ensemble model. (For the final model applied to the test data, the whole training set can be used.)**

Response to Comments c and d:

We thank the reviewer for the helpful suggestions. However, the PCs were chosen from the whole training set because we think that principal component analysis (PCA) is not a machine learning method, but a statistical technique for reducing the dimensionality of a dataset. It linearly transforms the data into a new coordinate system where most of its variation can be described with fewer dimensions than the initial data. Therefore, it may be more appropriate to treat the selection of PCs as data transformation of the training dataset instead of hyperparameter optimization.

- e. How were confidence intervals for AUCs computed?**

Author response:

We apologize for leaving out the calculation method of 95% CI. The CI of AUC was calculated with 2,000 stratified bootstrap replicates.

- 7. The term biomarker refers to some characteristic *evaluated* as an indicator of a medical state and is accurately and reproducibly measured. In the context of this manuscript, the term 'biomarker' used throughout is misleading. The authors should consider another term, such as 'feature', 'data modality', 'metric', 'measurement'.**

Author response:

We thank the reviewer for pointing this out. We have changed the term biomarker to "feature" or "modality" throughout the manuscript.

- 8. The study falls short in Figure 6, since the tissue-of-origin determination could likely be substantially improved. Instead of ATAC data, they should also use available methylation data from tissues, which would likely improve the accuracy of tissue-of-origin determination. Also, there should be a discussion on the modest performance of this analysis.**

Author response:

We agree with the reviewer that the performance of tissue-of-origin (TOO or CSO) classification remains to be improved and appreciate the reviewer's suggestion for using tissue methylation data. As mentioned in the manuscript, the THEMIS approach utilized shallow sequencing (~2X coverage) and had to use either broad genome segments or aggregate short regions for reliable quantification of methylation and fragmentation. Our TOO classifier adopted the latter strategy using cancer type specific accessible regulatory elements (most are TSS-distal enhancers) identified by ATAC-seq of TCGA cancer tissues. Although in general accessible regulatory elements tend to be hypomethylated, methylation profiles of these ATAC-seq peaks remain to be better characterized with whole-genome methylation sequencing datasets from multicancer tissues, which unfortunately are not available to the scientific community yet. Recently, a methylation atlas study of normal cell types has identified cell type specific loci and found that those unmethylated are often located in enhancers (Loyfer et al. 2023). By analyzing cancer tissue methylation data, we expect to not only supplement our TOO markers with differentially methylated genome regions other than regulatory elements but also refine the clustering of ATAC-seq peaks by methylation signals. We will conduct these investigations once tissue data becomes available.

We think that one of the main limits to our current TOO performance is the diversity of cancer types and etiologies included in the ATAC-seq study, which was discussed in the Discussion section. For example, as shown in Figure 6A, Cluster 9 (liver) was not significantly hypomethylated in LIHC plasma, which was in sharp contrast with Cluster 2 (colon) in COREAD plasma. This may suggest heterogeneity in the etiology of liver cancer between the Western and Chinese populations, driven by alcohol-consumption and HBV-infection respectively. We anticipate the availability of chromatin accessibility as well as methylation datasets from more diverse cancer types, subtypes, and etiologies to facilitate the refinement of our TOO markers and the improvement of prediction accuracy. We have modified the Discussion section to elaborate on differences in liver cancer etiology (Lines 529 - 531) and include model improvement strategy with tissue methylation data (Lines 544 - 549).

Reviewer #4 (Remarks to the Author): expertise in DNA methylation analysis

In this manuscript, Gao and colleagues proposed THEMIS, a machine learning approach for early cancer detection and classification from cell free DNA. The concept is not novel with Galleri, the commercial product available from GRAIL. The TET-APOBEC approach is also commercially available. Therefore, the major improvement has to be the improved performance (prediction accuracy). When taking at the face value, the reported accuracy does outperform that from GRAIL. However, there are many arbitrary design options in the model without careful justification and may lead into overfitting concerns.

Author response:

We thank the reviewer for pointing out concerns about novelty of this study and we apologize for insufficient clarifications of our concept advance in the original manuscript. We agree with the reviewer that early cancer detection and classification using cfDNA methylation has been commercially available from GRAIL and the TET-APOBEC approach for methylation sequencing (EM-seq) has also been commercially available from NEB. However, compared with Galleri using traditional and expensive methylation panels for deep targeted sequencing, THEMIS adopted a novel and cost-effective approach by applying EM-seq to cfDNA for whole-genome methylation profiling at low-depth. Nowadays, cfDNA methylomics and fragmentomics are becoming two popular mainstays for cancer applications and as pointed out by Reviewer 3, development of methodologies to extract the most tumor specific information is paramount in this field. This manuscript is the first proof-of-concept study to demonstrate the applicability of EM-seq based whole-methylome sequencing (WMS) for simultaneous analysis of cfDNA methylation, fragmentation, and CNA features for more powerful cancer detection, which introduces a novel approach to this fast-developing field. In addition, the WMS datasets generated in this study are of great value in comprehensive characterization of cfDNA epigenetic characteristics including but not limited to methylation and fragmentation, as recognized by other reviewers. We hope these discussions have addressed the reviewer's concern about the merit of this study, and our responses to overfitting concerns are provided below on a point-by-point basis.

Major comments:

1. For a AI model to be clinically useful, it needs to be robust (say from different processing lab etc). Although this manuscript reported a cohort from 6 different hospital, they were pooled together before randomly assigning 30% as testing. Therefore, there is no strong batch difference between the training dataset and the testing dataset. However, in clinical application, the sample was newly collected and may have latent batch difference that may affect the sample processing and data acquisition, which may in turn negatively influence the performance. Therefore, I will suggest the authors to validate their model in a new, independent cohort. Alternatively, they could select patients from 4 hospital as training and those from the remaining 2 hospitals as testing.

Author response:

We are very grateful for this suggestion on evaluating the effect of batch difference on model performance. We have performed the following two analyses to address this question:

I. As suggested in Comment #6 from Reviewer 3, we have added tSNE plots of each feature for healthy controls and each cancer type and indicated the hospital source in Figure S3. We observed no obvious batch effects among the six hospitals.

II. We have included an independent cohort to validate the performance of our approach based on availability, which comprises 475 high-risk noncancer controls and 263 patients of BRCA, ESCA, STAD, LIHC, and PACA, most of which are at early stage. Demographics of this cohort have been summarized in Table S1, and the AUCs (using noncancer individuals as controls) and sensitivities for each cancer type have been added in Figure S13. Not surprisingly, we have observed slightly decreased AUCs (0.900 overall) in the validation cohort probably for two main reasons. First, all the validating controls bear high-risk diseases and are older in age than the healthy controls used for model training. Second, the cases of this validation cohort are primarily early-stage patients. With the cutoff rendering 99% training specificity, the overall sensitivity of the validation cohort is 60%. Meanwhile, the specificity for the high-risk noncancer controls is 97.5%. Because we currently do not have complete one-year follow-up, we cannot rule out the possibility that some controls are actually cancer patients and thus overestimating the false positive rate. We understand that an independent validation cohort with similar sample composition as the training cohort can give a more accurate estimate of our model generalizability, but that would entail significant long-term efforts in patient recruitment, experiments, and data analysis. We hope that the results with the currently added validation cohort have demonstrated the adequate performance of our approach for clinical use wherein high specificity for noncancers (including high-risk diseases) and satisfactory sensitivity for early-stage cancers are desired. We have added the results of the validation cohort in the Results section (Lines 365 - 384).

2. There are many arbitrary design options throughout the model, which further boosts the overfitting suspicion. Here is just a few examples:

a. The authors reported hyperparameter tuning using 13 bootstraps. Why 13 instead of 10, 20 or 30?

Author response:

We apologize for this inaccurate description of the bootstrap method and have revised the text in the Methods section (Lines 693 - 695). In fact, we performed 10 bootstraps and added three additional bootstraps to include any samples left out in the original 10 bootstraps. We aimed to ensure that all samples were used in model training.

b. The normalization scheme is largely inconsistent. For example, no normalization for MFR, z-scoring for FSI and quantile normalization against

healthy controls for FEM. Why?

Author response:

We are sorry for insufficient descriptions of the rationales for the choice of data normalization methods which aimed to reduce batch effects while maintaining cancer signals. Because the cancer genome is characterized by widespread hypomethylation, we used raw MFR data after filtering fragments with a conversion rate below 95%. Unlike MFR, the composition of fragments in a library is highly sensitive to variations in library preparation and the ratios of short to long fragments (FSI) can be dramatically different among samples (Cristiano et al. 2019; Mathios et al. 2021). Therefore, z-score normalization was employed to make FSI values comparable between samples. Similarly, the frequencies of the 256 4-mer fragment end motifs are also subject to experimental batch effects and interdependent (i.e., the sum of the frequencies must be 1). Considering these, we utilized quantile normalization to maintain the ranking of FEM frequencies.

c. Different machine learning models were used for different models. Moreover, why not a single model that combines all features?

Author response:

We chose different machine learning methods for different data modalities based on their dimensionality relative to sample size. Please refer to Comment #6 from Reviewer 2 for our detailed explanations, which have also been added to the Methods section (Lines 719 - 722, 727 - 729, and 768 - 769).

Regarding the second question, we appreciate the reviewer's suggestion to combine all features in a single model and we looked into the matter during the model development. We so far shunned this approach because of several concerns. First, the dimensionality of individual feature is already quite high relative to our sample size and simply combining them would result in an even bigger *curse of dimensionality* to overcome. Our current approach to overcome the curse of dimensionality is to reduce the dimensionality for each data modality separately via PCA. For a single model combining all features, we so far have not figured out a sensible way to carry out dimensionality-reduction. Combining all raw features and performing a single PCA does not make sense because of the wildly different nature of each modality. On the other hand, selecting the optimal number of PCs for each modality first and then combining them introduces additional complexity as it is difficult to define "the optimal number" for each modality. In the end, we decided that the complementarity among the four modalities is better elucidated by running the appropriate machine learning method for each modality separately and combining the results in a second-tier model. However, we are actively investigating if a custom-designed deep-learning model would be able to combine all features and produce a better result.

REFERENCES

- Chan, K. C. Allen, Peiyong Jiang, Carol W. M. Chan, Kun Sun, John Wong, Edwin P. Hui, Stephen L. Chan, et al. 2013. "Noninvasive Detection of Cancer-Associated Genome-Wide Hypomethylation and Copy Number Aberrations by Plasma DNA Bisulfite Sequencing." *Proceedings of the National Academy of Sciences* 110 (47): 18761–68. <https://doi.org/10.1073/pnas.1313995110>.
- Chen, Han, and Han Liang. 2020. "A High-Resolution Map of Human Enhancer RNA Loci Characterizes Super-Enhancer Activities in Cancer." *Cancer Cell* 38 (5): 701–715.e5. <https://doi.org/10.1016/j.ccell.2020.08.020>.
- Cohen, Joshua D., Lu Li, Yuxuan Wang, Christopher Thoburn, Bahman Afsari, Ludmila Danilova, Christopher Douville, et al. 2018. "Detection and Localization of Surgically Resectable Cancers with a Multi-Analyte Blood Test." *Science* 359 (6378): 926–30. <https://doi.org/10.1126/science.aar3247>.
- Corces, M. Ryan, Jeffrey M. Granja, Shadi Shams, Bryan H. Louie, Jose A. Seoane, Wanding Zhou, Tiago C. Silva, et al. 2018. "The Chromatin Accessibility Landscape of Primary Human Cancers." *Science* 362 (6413). <https://doi.org/10.1126/science.aav1898>.
- Cristiano, Stephen, Alessandro Leal, Jillian Phallen, Jacob Fiksel, Vilmos Adleff, Daniel C. Bruhm, Sarah Østrup Jensen, et al. 2019. "Genome-Wide Cell-Free DNA Fragmentation in Patients with Cancer." *Nature* 570 (7761): 385–89. <https://doi.org/10.1038/s41586-019-1272-6>.
- Duda, R. O., and P. E. Hart. 1973. "Pattern Classification and Scene Analysis." 1973.
- Erger, Florian, Deborah Nörling, Domenica Borchert, Esther Leenen, Sandra Habbig, Michael S. Wiesener, Malte P. Bartram, et al. 2020. "CfNOMe — A Single Assay for Comprehensive Epigenetic Analyses of Cell-Free DNA." *Genome Medicine* 12 (1): 54. <https://doi.org/10.1186/s13073-020-00750-5>.
- Han, Diana S.C., Meng Ni, Rebecca W.Y. Chan, Vicken W.H. Chan, Kathy O. Lui, Rossa W.K. Chiu, and Y.M. Dennis Lo. 2020. "The Biology of Cell-Free DNA Fragmentation and the Roles of DNASE1, DNASE1L3, and DFFB." *The American Journal of Human Genetics* 106 (2): 202–14. <https://doi.org/10.1016/j.ajhg.2020.01.008>.
- Jamshidi, Arash, Minetta C. Liu, Eric A. Klein, Oliver Venn, Earl Hubbell, John F. Beausang, Samuel Gross, et al. 2022. "Evaluation of Cell-Free DNA Approaches for Multi-Cancer Early Detection." *Cancer Cell*, November, S153561082200513X. <https://doi.org/10.1016/j.ccell.2022.10.022>.
- Leary, Rebecca J., Mark Sausen, Isaac Kinde, Nickolas Papadopoulos, John D. Carpten, David Craig, Joyce O'Shaughnessy, et al. 2012. "Detection of Chromosomal Alterations in the Circulation of Cancer Patients with Whole-Genome Sequencing." *Science Translational Medicine* 4 (162). <https://doi.org/10.1126/scitranslmed.3004742>.
- Liu, M.C., G.R. Oxnard, E.A. Klein, C. Swanton, M.V. Seiden, Minetta C. Liu, Geoffrey R. Oxnard, et al. 2020. "Sensitive and Specific Multi-Cancer Detection and Localization Using Methylation Signatures in Cell-Free DNA." *Annals of Oncology* 31 (6): 745–59. <https://doi.org/10.1016/j.annonc.2020.02.011>.
- Loyfer, Netanel, Judith Magenheimer, Ayelet Peretz, Gordon Cann, Joerg Bredno, Agnes Klochender, Ilana Fox-Fisher, et al. 2023. "A DNA Methylation Atlas of Normal Human Cell Types." *Nature* 613 (7943): 355–64. <https://doi.org/10.1038/s41586-022-05580-6>.

- Mathios, Dimitrios, Jakob Sidenius Johansen, Stephen Cristiano, Jamie E. Medina, Jillian Phallen, Klaus R. Larsen, Daniel C. Bruhm, et al. 2021. "Detection and Characterization of Lung Cancer Using Cell-Free DNA Fragmentomes." *Nature Communications* 12 (1): 5060. <https://doi.org/10.1038/s41467-021-24994-w>.
- Robin, Xavier, Natacha Turck, Alexandre Hainard, Natalia Tiberti, Frédérique Lisacek, Jean-Charles Sanchez, and Markus Müller. 2011. "PROC: An Open-Source Package for R and S+ to Analyze and Compare ROC Curves." *BMC Bioinformatics* 12 (1): 77. <https://doi.org/10.1186/1471-2105-12-77>.
- Siejka-Zielińska, Paulina, Jingfei Cheng, Felix Jackson, Yibin Liu, Zahir Soonawalla, Srikanth Reddy, Michael Silva, et al. 2021. "Cell-Free DNA TAPS Provides Multimodal Information for Early Cancer Detection." *Science Advances* 7 (36). <https://doi.org/10.1126/SCIADV.ABH0534>.
- Ushijima, Toshikazu. 2005. "Detection and Interpretation of Altered Methylation Patterns in Cancer Cells." *Nature Reviews Cancer* 5 (3): 223–31. <https://doi.org/10.1038/nrc1571>.
- Zhang, Zhao, Joo-Hyung Lee, Hang Ruan, Youqiong Ye, Joanna Krakowiak, Qingsong Hu, Yu Xiang, et al. 2019. "Transcriptional Landscape and Clinical Utility of Enhancer RNAs for ERNA-Targeted Therapy in Cancer." *Nature Communications* 10 (1): 4562. <https://doi.org/10.1038/s41467-019-12543-5>.

REVIEWER COMMENTS

Reviewer #1 (Remarks to the Author):

The authors addressed most of my concerns. However, for the "contribution of each modality in the ensemble model". i can see that CAFF contribute most in the model, which is not surprised in late-stage cancer given most of their samples are late-stage samples. But what i really mean is that in the early-stage cancers, what is the major contributor?

2. given only ~30% samples are early-stage, i still don't think early-stage cancer detection is appropriate in the title...

Reviewer #2 (Remarks to the Author):

I appreciate the substantial efforts the authors have made to improve the manuscript based on the combined comments from all four reviewers. I think the manuscript has improved to the point it can be published.

The only remaining suggestion I have is to add 1-2 sentences on the intended use case (screening). The training of the model occurs on mostly cancer samples (of various grades), whereas in a screening setting one expect at most 1.5% 'positives' of presumably mostly low-grade. The PPV of 59%(which they do calculate and provide in the rebuttal) is not mentioned anywhere in the main text. I appreciate the response that this is a POC study, but this should be more concretely discussed / acknowledged in the manuscript (current modification is an improvement, but I would thus be more specific).

Reflection on comments of reviewer 3 and the author rebuttal:

Point 1: I think the authors adequately addressed this comment; but the authors should make clearer how they modified the manuscript to include the discussion in the rebuttal. The references to cfNOME for instance, require some discussion (it is only mentioned now, without any qualitative comparison, like in the rebuttal). The original point that the feature extraction is rudimentary is not really addressed, but to be honest, the fact that it works with simple approaches is also worth something. The data will be a major resource (but please note my critical remark on that below).

Point 2-4: OK

Point 5: Also raised by me. I think the analyses performed are adequate.

Point 6a: Adequate.

Point 6b-c: This is a bit of a tricky point. I think the reviewer is absolutely right to ask for including the PCA and selection within the Cross validation. I understand the boiler-plate answer from the authors, but I don't see why they just don't include the requested analysis (which is really straightforward). I agree with the reviewer that the authors should demonstrate the performance for a situation as closely resembling a truly 'new patient' as possible, i.e. meaning that this new datapoint can and should not influence the PCA or any other data transformation. My recommendation: ask for these analyses.

Point 7: OK

Point 8: The reviewer indeed points to a somewhat weaker analysis in the paper. It is a bit of an afterthought, and I definitely see many points for improvement for this analysis. The question is: would it be ok to leave open for the community to further address. This will be enabled by providing easy access to the data. I think the editor would need to make a judgement call here on how important it is to have this analysis top-notch. As it stands, it is ok, but there is certainly room for improvement (also after the revision).

One more critical remark:

I was unable to retrieve the data, check if the data was available and / or find out what would be required to get access to these data. A large part of the value of this work is to provide access to other researchers to the data. Besides, it would be essential to allow others to reproduce results. I advise the editor to strictly check that the data is available.

Reviewer #4 (Remarks to the Author):

In the revision manuscript, , Gao and colleagues have addressed some but not all concerns I raised during the first round of review. Specifically I still have concerns regarding the model generalizability and potential overfitting.

1) Batch effects: Essentially I am not arguing there are batch effects in the current data. Instead, I am suggesting that they need to establish that the current scheme is robust to potential batch effects in future, prospective application. In other words, the model could be quite useful if the authors can show that samples collected and processed by a third party /independent facility achieved a comparable performance. Otherwise, it would be just a fine-tuned model trained and used by the authors. Given the

lack of clear novelty of the classifier (sample source, processing approach and classifier scheme), the latter scenario is not very useful for the readers.

From the the study design and participants section, the “independent validation cohort” appeared to be from the same 6 hospitals and does not represent a potential batch effect that is different from the samples already included in the initial training.

A side note for the performance in the validation cohort: although I agree with the authors that they cannot rule out the possibility that some controls are actually cancer patients and thus overestimating the false positive rate (reported at 97.5%), the same logic argues that they are also OVERESTIMATING the sensitivity (reported at 60%).

2) The probability of any specific sample to be included in a single bootstrap is around 63.2%, which means the probability of excluding a specific sample in any run of the 10 bootstraps is $(1 - 0.632)^{10} = 4.55E-5$. The current study included 780 patients and 497 normal subjects and training was performed on 70% of the samples (894 subjects). Therefore, the probability of excluding at least one sample in all 10 bootstraps is less than $897 * 4.55E-5 = 0.04$. My concern here is whether the given rationale for the additional 3 bootstraps (to include all samples in training) is justified?

3) Normalization: The authors argue the rationale to normalize FSI and FEM, but not MFR. However, the justification to different normalization schemes for FSI and FEM is elusive at best. The choice of normalization schemes is still largely arbitrary to me. What happens if we run quantile normalization for FEM and z-score for FSI?

4) The number of features (53/42/31 for MFR/FSI/FEM, respectively) is modest for OMICs-era machine learning approaches, given that the authors have 894 samples for training. Therefore, there are likely additional reasons beyond curse of dimensionality behind the poor performance when they combine all features together in a single model.

Dear reviewers,

We sincerely appreciate all the detailed and constructive comments provided by the reviewers. Based on the comments combined, we have made modifications to our original modeling methods including: (I) For FEM modeling, we have switched from quantile-normalized to raw motif frequency to reduce overfitting; (II) For the training performance of MFR/FSI/FEM, we have included principal component analysis (PCA) and the selection of principal components (PCs) into cross validation instead of using the whole training set; (III) The number of bootstraps for MFR/FSI/FEM modeling has been revised to 10; (IV) We have performed log₁₀ transformation of CAFF PA scores before the stage-2 ensemble model to control the distribution of CAFF values. We think that these modifications have improved our model robustness and have updated the results of the new models throughout the revised manuscript. We have also provided detailed point-by-point responses to reviewer comments below.

Reviewer #1 (Remarks to the Author):

1. The authors addressed most of my concerns. However, for the "contribution of each modality in the ensemble model". i can see that CAFF contribute most in the model, which is not surprised in late-stage cancer given most of their samples are late-stage samples. But what i really mean is that in the early-stage cancers, what is the major contributor?

Author response:

We appreciate the reviewer's clarification and agree that deciphering the major contributor for the detection of early-stage cancer is an important question in this field. To address this question, we stratified cancer samples by stage and trained an early-stage (I and II) model and a late-stage (III and IV) model with healthy controls, respectively. It is to be noted that we have updated data pre-processing for some models, including the use of raw motif frequencies for FEM modality and log-transformation of CAFF PA scores for the ensemble model, as described in the beginning of this "response to reviewers" letter. The coefficients for the early-stage and late-stage models as well as the all-stage model are listed in the attached Table 1. We find that FEM and MFR are the major contributors for early-stage samples. All modalities contribute to late-stage samples with FEM and FSI contributing more than MFR and CAFF. CAFF appears to be a less important feature especially in early stages, consistent with the notion that chromosome aneuploidy is not a sensitive circulating cancer biomarker for early detection.

Table 1. Coefficients of the ensemble elastic-net model for integration of modalities.

Predictor	Model coefficient		
	Early-stage	Late-stage	All-stage
Intercept	-0.73	-0.32	0.57
MFR	0.46	0.31	0.33
FSI	0.09	0.41	0.34
CAFF	0.00	0.15	0.06
FEM	0.71	0.53	0.58

2. given only ~30% samples are early-stage, i still don't think early-stage cancer detection is appropriate in the title...

Author response:

We understand the reviewer's concern about the performance of our approach for early-stage cancer detection given the relatively limited sample size of this proof-of-concept study. We have removed the term "early" from the title. The real-world detection power of our approach for early-stage cancers will be investigated in future large-scale prospective studies.

Reviewer #2 (Remarks to the Author):

I appreciate the substantial efforts the authors have made to improve the manuscript based on the combined comments from all four reviewers. I think the manuscript has improved to the point it can be published.

The only remaining suggestion I have is to add 1-2 sentences on the intended use case (screening). The training of the model occurs on mostly cancer samples (of various grades), whereas in a screening setting one expect at most 1.5% 'positives' of presumably mostly low-grade. The PPV of 59%(which they do calculate and provide in the rebuttal) is not mentioned anywhere in the main text. I appreciate the response that this is a POC study, but this should be more concretely discussed / acknowledged in the manuscript (current modification is an improvement, but I would thus be more specific).

Author response:

We thank the reviewer for the acknowledgement of our work and the suggestion to include the intended use in the manuscript. We have added descriptions of the intended use for cancer screening and the putative PPV in the Discussion section (Lines 499-509).

Reflection on comments of reviewer 3 and the author rebuttal:

Point 1: I think the authors adequately addressed this comment; but the authors should make clearer how they modified the manuscript to include the discussion in the rebuttal. The references to cfNOME for instance, require some discussion (it is only mentioned now, without any qualitative comparison, like in the rebuttal). The original point that the feature extraction is rudimentary is not really addressed, but to be honest, the fact that it works with simple approaches is also worth something. The data will be a major resource (but please note my critical remark on that below).

Author response:

We appreciate the reviewer's suggestion to include comparison with cfNOME in the manuscript. We have elaborated on the comparison with cfNOME in terms of the fraction of circulating renal versus tumor derived cfDNA as well as the required sequencing depth by the two methods in the Discussion section (Lines 470-478), which were discussed in the previous response letter.

We also agree with the reviewer that the current extraction of genomic features on a global scale is rudimentary, but the fact that it achieves satisfactory performance for multicancer detection may suggest the validity as well as the universality of these established cancer biomarkers. This leads us to reason that our approaches may also work for other cancer types, although it is beyond the scope of the current manuscript to test this hypothesis. To improve detection performance, we anticipate to refine feature extraction in the future by identification of *bona fide* cancer-relevant modality and genomic elements.

Finally, we apologize for not providing clearer instructions for the reviewer to retrieve the data. We deposited all WMS bam files in GSA for Human with accession no. HRA003209, and have now also deposited all fastq files under the same accession no. All sequencing files can be publicly accessed via the URL (<https://ngdc.cncb.ac.cn/gsa-human/s/Tp80zqDh>).

Point 2-4: OK

Point 5: Also raised by me. I think the analyses performed are adequate.

Point 6a: Adequate.

Point 6b-c: This is a bit of a tricky point. I think the reviewer is absolutely right to ask for including the PCA and selection within the Cross validation. I understand the boiler-plate answer from the authors, but I don't see why they just don't include the requested analysis (which is really straightforward). I agree with the reviewer that the authors should demonstrate the performance for a situation as closely resembling a truly 'new patient' as possible, i.e. meaning that this new datapoint can and should not influence the PCA or any other data transformation. My recommendation: ask for these analyses.

Author response:

We agree with the reviewer that for training performance the optimal strategy is to include PCA and PC selection within cross validation instead of using the whole training set. We have revised our MFR/FSI/FEM model to perform PCA and PC selection with the training folds (which are randomly selected 70% of the training cohort) and validate the resulting model with the validation folds (which are the remaining 30% of the training cohort) for each

bootstrap. To evaluate the influence of PC selection on model performance, we have trained our models with a range of top PCs explaining from 82% to 99% of data variance for each modality with 10 bootstraps. We have plotted the AUCs of the training and validation folds for all bootstraps in Supplementary Figure 16. We find that both training and validation AUCs are consistent among the bootstraps and the optimal variances appear to be 95%, 90%, and 95% for MFR, FSI, and FEM respectively to yield high validation AUCs and trivial overfitting. These results demonstrate the generalizability of our data transformation methods to new datasets and we have added these analyses to the description of model development in the Methods section (Lines 702-734).

Point 7: OK

Point 8: The reviewer indeed points to a somewhat weaker analysis in the paper. It is a bit of an afterthought, and I definitely see many points for improvement for this analysis. The question is: would it be ok to leave open for the community to further address. This will be enabled by providing easy access to the data. I think the editor would need to make a judgement call here on how important it is to have this analysis top-notch. As it stands, it is ok, but there is certainly room for improvement (also after the revision).

Author response:

We agree with the reviewer that the TOO/CSO performance of the current manuscript remains to be improved. We anticipate to refine our TOO/CSO marker selection once more cancer tissue ATAC-seq and methylation data become available, as discussed in the previous response letter.

One more critical remark:

I was unable to retrieve the data, check if the data was available and / or find out what would be required to get access to these data. A large part of the value of this work is to provide access to other researchers to the data. Besides, it would be essential to allow others to reproduce results. I advise the editor to strictly check that the data is available.

Author response:

Please refer to the response to Point 1 for data access.

Reviewer #4 (Remarks to the Author):

In the revision manuscript, Gao and colleagues have addressed some but not all concerns I raised during the first round of review. Specifically I still have concerns regarding the model generalizability and potential overfitting.

1) Batch effects: Essentially I am not arguing there are batch effects in the current data. Instead, I am suggesting that they need to establish that the current scheme is robust to potential batch effects in future, prospective application. In other words, the model could be quite useful if the authors can show that samples collected and

processed by a third party /independent facility achieved a comparable performance. Otherwise, it would be just a fine-tuned model trained and used by the authors. Given the lack of clear novelty of the classifier (sample source, processing approach and classifier scheme), the latter scenario is not very useful for the readers.

From the the study design and participants section, the “independent validation cohort” appeared to be from the same 6 hospitals and does not represent a potential batch effect that is different from the samples already included in the initial training.

Author response:

We appreciate the reviewer's clarification and totally agree with the reviewer on the importance of generalizability for machine learning models to be useful. Indeed, the validation cohort was recruited from the same 6 hospitals after the THEMIS model had been developed and locked with the MONITOR cohort. Therefore, it represented other potential batch effects (e.g., sample collection time, reagent lots, etc.) except hospital source. Ideally, our model should be tested on external datasets generated by a third party, but to our knowledge so far there are no cancer cfDNA enzymatic methylation sequencing datasets available from other studies. It is neither feasible to obtain ethical approval and recruit enough participants from a new hospital within the timeframe of the revision for the current manuscript. To address the question of potential hospital batch effect, we have adopted an alternative approach as suggested by the reviewer in the first round of review to evaluate the impact of hospital batch effect on model performance. Specifically, because in our data all cancer (except BRCA) and healthy cohorts were recruited from two to four hospitals (see Table S6 and the attached Table 2 for details), to evaluate hospital batch effect we split the samples by hospital: for each non-BRCA cancer/healthy cohort, samples from one hospital are used for model testing and samples from the remaining hospitals for model training. This scheme yields a total of 384 split-by-hospital combinations. We have implemented feature extraction, trained the model for each combination, and summarized model performance in Supplementary Figure 13. We observe comparable AUCs among the 384 split-by-hospital combinations for individual modalities and the integrative model (Figure S13A), which demonstrates little hospital batch effect.

However, we find that model performance appears to be associated with the proportion of early-stage (I and II) cancer patients in each combination (Figure S13B), which makes sense as the cancer detection power is dependent on ctDNA fraction in the blood.

We have added these results to the Results section (Lines 318-386) and Figure S13. We have also removed the results of the independent validation cohort since it could not address the reviewer's concern of hospital batch effect.

Table 2. Hospital source of the MONITOR cohort samples.

Sample type	Hospital #					
	1	2	3	4	5	6
BRCA	66					
COREAD		30			36	84
ESCA			17	44		
HEALTHY		235	262			
LIHC	70					43
NSCLC		23	52	45	37	
PACA			75			44
STAD					42	72

A side note for the performance in the validation cohort: although I agree with the authors that they cannot rule out the possibility that some controls are actually cancer patients and thus overestimating the false positive rate (reported at 97.5%), the same logic argues that they are also OVERESTIMATING the sensitivity (reported at 60%).

Author response:

We understand the reviewer's concern about overestimate of the sensitivity due to incomplete follow-up of non-cancer controls. However, this should not be a concern for our retrospectively analyzed validation cohort because our case-control study was constructed to evaluate the performance of our approach in detecting cancer signals from known cancer samples instead of a screening population of mostly healthy individuals. That being said, the validation cohort was employed to solely evaluate the generalizability of our model to external datasets that were not included in model development, and thus the real-world performance (including sensitivity, specificity, PPV, etc.) of our model remains to be tested in a prospective screening study with long-term follow-up. These limitations of our current case-control study are discussed in the Discussion section (Lines 573-577)

2) The probability of any specific sample to be included in a single bootstrap is around 63.2%, which means the probability of excluding a specific sample in any run of the 10 bootstraps is $(1-0.632)^{10}=4.55E-5$. The current study included 780 patients and 497 normal subjects and training was performed on 70% of the samples (894 subjects). Therefore, the probability of excluding at least one sample in all 10 bootstraps is less than $897*4.55E-5=0.04$. My concern here is whether the given rationale for the additional 3 bootstraps (to include all samples in training) is justified?

Author response:

We sincerely appreciate the reviewer's careful calculation and totally agree that the probability of excluding a sample in all 10 bootstraps is so minuscule that model

performance should not be affected. Indeed, we have re-trained the models without the additional three bootstraps and obtained highly similar results as the original 13 bootstraps. We have thus removed the three additional bootstraps from our model training procedure and modified the Methods section to reflect this revision. Again, we thank the reviewer for this valuable suggestion.

3) Normalization: The authors argue the rationale to normalize FSI and FEM, but not MFR. However, the justification to different normalization schemes for FSI and FEM is elusive at best. The choice of normalization schemes is still largely arbitrary to me. What happens if we run quantile normalization for FEM and z-score for FSI?

Author response:

We thank the reviewer for looking into this important question of data normalization and proposing investigation of alternative normalization methods. We originally ran quantile normalization for FEM and z-score for FSI in the last version of the manuscript, so we assume that the reviewer was suggesting the opposite normalization scheme in the comment (i.e., quantile normalization for FSI and z-score for FEM). To investigate the impact of normalization methods on model performance, we have trained FSI and FEM modalities with three different data preprocessing methods (no normalization, quantile normalization, and z-score) respectively, and model AUCs are listed in the attached Table 3.

For FSI, the level of short to long fragment ratio is highly sample-specific as shown in the attached Figure 1, which depicts the genome-wide FSI profiles of 497 healthy individuals in the MONITOR cohort. After normalization by z-score or quantile normalization across the 502 genome bins for each sample, these healthy controls share similar profiles. Indeed, the top principle component (PC) can explain over 99% variance of raw FSI data. By contrast, to explain at least 90% of data variance, 42 PCs are needed for normalized FSI data by z-score or quantile normalization. Both training and testing AUCs are similar between these two normalization methods, suggesting that both methods work for FSI. We prefer z-score because it is a simpler scheme that does not require a group of samples, which was also employed by the original publication of this circulating cancer feature (Cristiano et al., *Nature*, 2019).

Unlike FSI, FEM modality with raw motif frequency demonstrates slightly better testing AUC than preprocessing by either quantile normalization or z-score. This result suggests that the signal-to-noise ratio of FEM may be weakened by the above two normalization methods. Therefore, we have revised our FEM methods in the manuscript by using raw motif frequencies for modeling.

Table 3. Model AUCs with different data preprocessing methods.

FSI			FEM		
Normalization method	Training AUC	Testing AUC	Normalization method	Training AUC	Testing AUC
No normalization	NA	NA	No normalization	0.951	0.934
Quantile normalization	0.901	0.908	Quantile normalization	0.968	0.926
Z-score	0.918	0.909	Z-score	0.932	0.927

Figure 1. FSI profiles of 497 healthy individuals along the genome.

4) The number of features (53/42/31 for MFR/FSI/FEM, respectively) is modest for OMICs-era machine learning approaches, given that the authors have 894 samples for training. Therefore, there are likely additional reasons beyond curse of dimensionality behind the poor performance when they combine all features together in a single model.

Author response:

We thank the reviewer for providing a suggestion to integrate modalities by concatenating PCs for one-stage modeling. In our response to the first round of review, we mentioned "curse of dimensionality" to describe the large number of raw features for MFR (1,846 1-Mb windows), FSI (502 5-Mb windows), and FEM (256 4-mer motifs). To reduce data dimensionality, we performed PCA and in the current revised manuscript we selected top PCs explaining 95%, 90%, and 95% of data variance for MFR, FSI, and FEM, which included 53, 42, and 21 top PCs respectively. To assess the performance of one-stage modeling with the three modalities, we have concatenated the 116 PCs for each sample as suggested by the reviewer and applied different machine learning models including logistic regression (LR), random forest (RF), and linear support vector machine (LinearSVM) with bootstraps. Training versus testing AUCs suggest the presence of various degrees of overfitting for all these models (LR: 0.985 vs 0.961; RF: 1.000 vs 0.971; and LinearSVM: 0.979 vs 0.957). By contrast, integration of the three modalities in the two-stage modeling approach shows less overfitting (training AUC 0.971 vs testing AUC 0.965). Again, we sincerely appreciate the reviewer's suggestion of alternative modelling and totally agree that combining all features together in a single model is a feasible and promising direction. In follow-up studies, we aim to thoroughly characterize each modality and find the optimal approach to feature integration.

REVIEWERS' COMMENTS

Reviewer #1 (Remarks to the Author):

the authors have addressed all of my concerns now.

Reviewer #2 (Remarks to the Author):

I have not further comments

Reviewer #4 (Remarks to the Author):

The authors mostly addressed my concerns.

Just one minor comment: the small drop of testing performance (compared to training) is normal. The random forest model using combined PCs outperformed the reported THEMIS model. They may consider reporting this 1-step RF model as the final published model.

Reviewer #1 (Remarks to the Author):

The authors have addressed all of my concerns now.

Reviewer #2 (Remarks to the Author):

I have not further comments

Reviewer #4 (Remarks to the Author):

The authors mostly addressed my concerns. Just one minor comment: the small drop of testing performance (compared to training) is normal. The random forest model using combined PCs outperformed the reported THEMIS model. They may consider reporting this 1-step RF model as the final published model.

Author response:

We appreciate the reviewer's suggestion for using the 1-step RF model instead of the 2-step THEMIS model. However, despite demonstrating slightly higher testing AUC (RF 0.971 vs THEMIS 0.966), the larger gap between the training and testing AUC for the RF model ($=0.029$) suggests it is more overfitting than the THEMIS model (AUC gap= 0.005). We plotted the distributions of the prediction probability scores for the two models stratified by healthy and cancer stages in the attached Figure 1. The RF model displays much larger discrepancy in terms of prediction scores between the training cohort and the testing cohort in both healthy controls and varying-stage cancers, which casts doubt on its generalizability. Indeed, after we derived a score cutoff at 99% specificity in the training cohort and applied the cutoff to the testing cohort for both models, the RF model displayed an abnormal pattern in the sensitivities across cancers (attached Figure 2A). In another word, for the RF model, its cutoff value determined through the training cohort is inapplicable to the testing cohort. In contrast, the prediction scores of the THEMIS model are more consistent between the training and the testing cohorts for both healthy controls and cancer samples (attached Figures 1 and 2B), demonstrating its much better generalizability. Therefore, we prefer to publish the 2-stage THEMIS model as the final model considering its better generalizability, which is crucial for future real-world clinical applications.

Figure 1. Box plots depicting prediction probability scores of the THEMIS model and the random forest model. Samples are stratified by healthy scores and cancer stage. Wilcoxon test is applied to calculate p-values (ns: $p > 0.05$; *: $p \leq 0.05$; **: $p \leq 0.01$; ***: $p \leq 0.001$; ****: $p \leq 0.0001$).

Figure 2. Detection sensitivities of the random forest model and the THEMIS model. Sensitivity of individual cancer types by clinical stage at a training specificity of 99% is depicted with 95% Wilson confidence interval. The number of samples in the training and the test cohort (separated by a vertical line) are indicated below the plot. Cancer samples with unknown stages are omitted from display.